# Deep reconstruction of strange attractors from time series

**William Gilpin**[*]
Quantitative Biology Initiative
Harvard University
Cambridge, MA 02138
`wgilpin@fas.harvard.edu`

## Abstract

Experimental measurements of physical systems often have a limited number of independent channels, causing essential dynamical variables to remain unobserved. However, many popular methods for unsupervised inference of latent dynamics from experimental data implicitly assume that the measurements have higher intrinsic dimensionality than the underlying system—making coordinate identification a dimensionality reduction problem. Here, we study the opposite limit, in which hidden governing coordinates must be inferred from only a low-dimensional time series of measurements. Inspired by classical analysis techniques for partial observations of chaotic attractors, we introduce a general embedding technique for univariate and multivariate time series, consisting of an autoencoder trained with a novel latent-space loss function. We show that our technique reconstructs the strange attractors of synthetic and real-world systems better than existing techniques, and that it creates consistent, predictive representations of even stochastic systems. We conclude by using our technique to discover dynamical attractors in diverse systems such as patient electrocardiograms, household electricity usage, neural spiking, and eruptions of the Old Faithful geyser—demonstrating diverse applications of our technique for exploratory data analysis.

## 1 Introduction

Faced with an unfamiliar experimental system, it is often impossible to know *a priori* which quantities to measure in order to gain insight into the system's dynamics. Instead, one typically must rely on whichever measurements are readily observable or technically feasible, resulting in partial measurements that fail to fully describe a system's important properties. These hidden variables seemingly preclude model building, yet history provides many compelling counterexamples of mechanistic insight emerging from simple measurements—from Shaw's inference of the strange attractor driving an irregularly-dripping faucet, to Winfree's discovery of toroidal geometry in the *Drosophila* developmental clock [1, 2].

Here, we consider this problem in the context of recent advances in unsupervised learning, which have been applied to the broad problem of discovering dynamical models directly from experimental data. Given high-dimensional observations of an experimental system, various algorithms can be used to extract latent coordinates that are either time-evolved through empirical operators or fit directly to differential equations [3, 4, 5, 6, 7, 8, 9, 10]. This process represents an empirical analogue of the traditional model-building approach of physics, in which approximate mean-field or coarse-grained dynamical variables are inferred from first principles, and then used as independent coordinates in a reduced-order model [11, 12]. However, many such techniques implicitly assume that the

---

[*]Code available at: https://github.com/williamgilpin/fnn

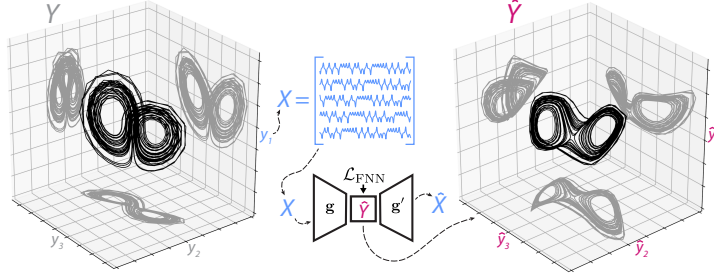

Figure 1: Overview of problem and approach. A univariate time series $y_1(t)$ is observed from a multivariate attractor $Y = [y_1(t)\ y_2(t)\ y_3(t)]$. This signal is converted into a time-lagged Hankel matrix $X$, which is used to train an autoencoder with the false-nearest-neighbor loss $\mathcal{L}_{\text{FNN}}$. The latent variables reconstruct the original coordinates.

degrees of freedom in the raw data span the system's full dynamics, making dynamical inference a dimensionality reduction problem.

Here, we study the inverse problem: given a single, time-resolved measurement of a complex dynamical system, is it possible to reconstruct the higher-dimensional process driving the dynamics? This process, known as state space reconstruction, is the focus of many classical results in nonlinear dynamics theory, which demonstrate various heuristics for reconstructing effective coordinates given the time history of a system [13, 14]. Such techniques have broad application throughout the natural sciences, particularly in areas in which simultaneous multidimensional measurements are difficult to obtain—such as ecology, physiology, and climate science [15, 16, 17, 18]. However, these embedding techniques are strongly sensitive to hyperparameter choice, system dimensionality, non-stationarity, continuous spectra, and experimental measurement error—therefore requiring extensive tuning and in-sample cross-validation before they can be applied to a new dataset [19, 20, 15]. Additionally, current methods cannot consistently infer the underlying dimensionality as the original system, making them prone to redundancy and overfitting [21]. Several of these shortcomings may be addressable by revisiting classical techniques with contemporary methods, thus motivating our study.

Here, we introduce a general method for reconstructing the $d$-dimensional attractor of an unknown dynamical system, given only a univariate measurement time series. We introduce a custom loss function and regularizer, the false-nearest-neighbor loss, that allows recurrent autoencoder networks to successfully reconstruct unseen dynamical variables from time series. We embed a variety of dynamical systems, and we formalize several existing and novel metrics for comparing an inferred attractor to a system's original attractor—and we demonstrate that our method outperforms baseline state space reconstruction methods. We test the consistency of our technique on stochastic dynamical systems, and find that it generates robust embeddings that can effectively forecast the dynamics at long time horizons, in contrast with previous methods. We conclude by performing exploratory analysis of datasets that have previously been hypothesized to occupy strange attractors, and discover underlying attractors in systems spanning earth science, neuroscience, and physiology.

## 2    Background and Definitions

Suppose that a $d$-dimensional dynamical system $\dot{\mathbf{y}} = \mathbf{f}(\mathbf{y}, t)$ occupies an attractor $A$. The time-evolving state variable $\mathbf{y}$ may be represented abstractly by composition with a flow operator, $\mathbf{y}(t) = \mathcal{F} \circ \mathbf{y}(t_0)$. At any given instant in time, a measurement $\mathbf{x}(t)$ corresponds to composition with the operator, $\mathcal{M}$, such that $\mathbf{x}(t) = \mathcal{M} \circ \mathbf{y}(t) = \mathcal{M} \circ (\mathcal{F} \circ \mathbf{y}(t_0))$, where $d_m \equiv \dim \mathbf{x}_t$. We define the data matrix $X = [\mathbf{x}_1^\top\ \mathbf{x}_2^\top\ \cdots\ \mathbf{x}_N^\top]^\top$ as a collection of $N$ evenly-spaced measurements with timestep $\Delta t$. Many standard unsupervised embedding techniques for dynamical systems, such as proper orthogonal decomposition or dynamic mode decomposition, implicitly require that $d_m$ is sufficiently large that the measurement operator's basis spans that of the original system, $\text{span}(\mathcal{M}) \geq \text{span}(\mathcal{F})$ [5, 22]. This condition makes it possible to infer $A$ with sufficient measurements.

Here, we consider the case where high-dimensional time-resolved measurements are unavailable, $\text{span}(\mathcal{M}) < \text{span}(\mathcal{F})$, making it more challenging to infer the underlying dynamics. A common

example is the univariate case $d_m = 1$, such that $X = [x_1 \ x_2 \ \cdots \ x_N]^\top$. A standard solution in time series analysis is to augment the dimensionality of the measurements via the method of lags, in which the $T$ previous measurements are appended to each timestep to produce a multidimensional surrogate measurement $\mathbf{x}_i = [x_{i-T} \ x_{i-T+1} \ \cdots \ x_i]$. In principle, $T$ should be large enough that $x$ (and potentially $\mathbf{y}$) undergoes sufficient variation to provide information about the dynamics of each component $y_j$ of the underlying system. After augmenting dimensionality with lags, the measurement matrix $X \in \mathbb{R}^{N \times T}$ has Hankel structure along its diagonals, and here it will serve as the input for an unsupervised learning problem:

We seek a parametric similarity transformation $\hat{\mathbf{y}} = \mathbf{g}(\mathbf{x})$ such that $\hat{Y} \sim Y$, where $Y \in \mathbb{R}^{N \times d}$ and $\hat{Y} \in \mathbb{R}^{N \times d_E}$. The point set $Y = [\mathbf{y}_1^\top \ \mathbf{y}_2^\top \ \cdots \ \mathbf{y}_N^\top]^\top$ corresponds to a finite-duration sample from the true attractor $A$, and the point set $\hat{Y} = [\hat{\mathbf{y}}_1^\top \ \hat{\mathbf{y}}_2^\top \ \cdots \ \hat{\mathbf{y}}_N^\top]^\top$ refers to the embedding of $\mathbf{x}$ at the same timepoints. Because $Y$ (and thus $d$) is unknown, the embedding dimension hyperparameter $d_E$ imposes the number of coordinates in the embedding. For this reason, we seek similarity $\hat{Y} \sim Y$, rather than $\hat{Y} = Y$; the stronger condition of congruency $\hat{Y} \cong \hat{Y}$ cannot be assured because a univariate measurement series lacks information about the relative symmetry, chirality, or scaling of $Y$. This can be understood by considering the case where the measurement $\mathbf{x}$ corresponds to a projection of the dynamics onto a single axis, a process that discards information about the relative ordering of the original coordinates.

For general dynamical systems, the embedding function $\mathbf{g}$ satisfies several properties. For many dynamical systems, the attractor $A$ is a manifold with fractal dimension $d_F \leq d$ (a non-integer for many chaotic systems), which can be estimated from $Y$ using box-counting or related methods. Likewise, $\dim \hat{\mathbf{y}} = d_E$ is usually chosen to be sufficiently large that some embedding coordinates are linearly dependent, and so the intrinsic manifold dimension of the embedded attractor $E$ is less than or equal to $d_E$. Under weak assumptions on $\mathbf{f}$ and $\mathcal{M}$, the Whitney embedding theorem states that any such dynamical attractor $A$ can be continuously and invertibly mapped to a $d_E$-dimensional embedding $E$ as long as $d_E > 2d_F$. This condition ensures that structural properties of the attractor relevant to the dynamics of the system (and thus to prediction and characterization) will be retained in the embedded attractor as long as $d_E$ is sufficiently large.

However, while the Whitney embedding theorem affirms the feasibility of attractor reconstruction, it does not prescribe a specific method for finding $\mathbf{g}$ from arbitrary time series. In practice, $\mathbf{g}$ is often constructed using the method of delays, in which the embedded coordinates comprise a finite number of time-lagged coordinates, $\mathbf{g}(\mathbf{x}_i) = [x_{i-d_E\tau} \ x_{i-(d_E-1)\tau} \ \cdots \ x_i]^\top$. This technique was first applied to experimental data in the context of turbulence and other dissipative dynamical systems [23], and it is formalized by Takens' theorem, a corollary of the Whitney theorem that states that $\hat{Y}$ will be diffeomorphic to $Y$ for *any* choice of lag time $\tau$ [24]. However, the properties of $\hat{Y}$ strongly vary with the choice of lag time $\tau$ [14]. Additional theoretical and empirical studies with lagged coordinates suggest that, for many classes of measurement operations $\mathcal{M}$, it may be possible to construct embeddings $\hat{Y}$ that are not only diffeomorphic, but also isometric in the sense of preserving local neighborhoods around points on an attractor [25, 26, 27, 28, 29]—a property implicitly required for forecasting and dynamical analysis of reconstructed attractors [15]. This has led some authors to speculate that certain embedding techniques satisfy the Nash embedding theorem, a strengthening of the Whitney theorem that gives conditions under which an embedding becomes isometric for sufficiently large $d_E$ [30, 31].

## 3 Related Work

State space reconstruction with lagged coordinates is widely used in fields ranging from ecology, to medicine, to meteorology [32, 14, 33, 15, 17]. Many contemporary applications use classical methods for determining $\tau$ and $d_E$ for these embeddings [34, 13], although recent advances have helped reduce the method's sensitivity to these hyperparameters [35, 21]. Other works have explored the use of multiscale time lags inferred via information theoretic [36] or topological [37] considerations. However, in the presence of noise, time lagged embeddings may generalize poorly to unseen data, thus requiring extensive cross-validation and Bayesian model selection to ensure robustness [19, 38].

Embeddings may also be constructed via singular-value decomposition of the Hankel matrix, producing a set of "eigen-time-delay coordinates" [39, 40, 41]. These have recently been used to construct

high-dimensional linear operators that can evolve the underlying dynamics [42]. Other methods of constructing $\mathbf{g}(.)$ include time-delayed independent components [43], Laplacian eigenmaps [44, 45], Laplacian spectral analysis [46], and reservoir computers [47]. Kernel methods for time series also implicitly lift the time series into a higher-dimensional state space [26], either using fixed nonlinear kernels [48, 49, 50], or trainable kernels comprising small neural networks [51]. Several recent studies learn embeddings $\mathbf{g}(.)$ using an autoencoder, an approach we will revisit here [52, 53, 3, 54, 55, 56]. Variational autoencoders may be used to model the latent dynamics probabilistically [57, 58, 59], in which case rank penalties can enforce dynamical sparsity [4, 60, 7].

Here, we are particularly interested in the related, but distinct, problem of finding time series embeddings that most closely approximate the true dynamical attractor of the underlying system. Accordingly, we seek coordinates that are both predictive and parsimonious, which we quantify with a variety of similarity metrics described below. Our general approach consists of training a stacked autoencoder on the Hankel matrix of the system via a novel, sparsity-promoting latent-space regularizer, which seeks $d_E \approx d$.

## 4 Methods

### 4.1 Approach

We create physically-informative attractors from time series by training an autoencoder, a type of neural network generally used for unsupervised learning [61]. The encoder portion of the network $\mathbf{g}$ takes as input the Hankel measurement matrix $X$ (or a batch comprising a random subset of its rows), and it acts separately on each row to produce an estimate of the attractor $\hat{Y} = \mathbf{g}(X)$, which comprises the latent space of the autoencoder. The decoder $\mathbf{g}'$ takes $\hat{Y}$ as input, and attempts to reconstruct the input $\hat{X} = \mathbf{g}'(\hat{Y})$. The encoder and encoder are trained together

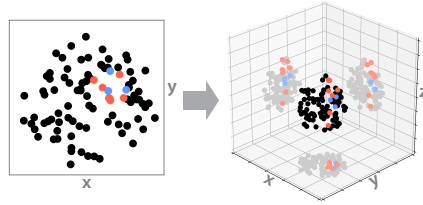

Figure 2: In a 2D projection of a 3D point cloud, false neighbors (red) separate when the system is lifted to 3D.

by composing $\hat{X} = \mathbf{g}'(\mathbf{g}(X))$, and minimizing the difference between $X$ and $\hat{X}$. After training the autoencoder, an embedding may be generated either from the training data or unseen test data using only the encoder, $\hat{Y} = \mathbf{g}(X)$. To constrain $\mathbf{g}'$ and $\mathbf{g}$, in addition to the reconstruction loss we introduce a novel sparsity-promoting loss $\mathcal{L}_{\text{FNN}}$, which functions as a latent-space activity regularizer.

We outline our approach in Figure 1. In our autoencoder, the hyperparameter $T$ specifies the width of the Hankel matrix (number of time lags) used as input features, while the hyperparameter $L$ (number of latent units) specifies the number of embedding coodinates. However, because of the regularizer, not all $L$ latent units will necessarily remain active after training; our autoencoder thus learns an effective embedding dimension $d_E \leq L$. Rather than culling low-activity latent units to produce an integer $d_E$, we define $d_E$ from our embedding $\hat{Y}$ post-training as a continuous quantity related to the distribution of relative variance across the $L$ latent units (this is similar to estimating the dimensionality of a PCA embedding from the dropoff in singular values). We aim to have the learned attractor $\hat{Y}$ match the unseen original attractor $Y$ as closely as possible, including $d_E \approx d$.

Our loss function $\mathcal{L}_{\text{FNN}}$ represents a variational formulation of the false-nearest-neighbors method, a popular heuristic for determining the appropriate embedding dimension $d_E$ when using the method of lags [13]. The intuition behind the technique is that a $d$-dimensional embedding with too few dimensions will have many overlapping points, which undergo large separation when the embedding is lifted to $d + 1$. These "false neighbors" only co-localize in $d$ dimensions due to having overlapping projections (Figure 2). The traditional false-nearest-neighbors technique asserts that the optimal embedding dimension $d_E$ occurs when the fraction of false nearest neighbors first approaches zero as $d$ increases. Here, we modify the technique to apply it iteratively during training: at each optimization step, the false neighbors fraction is estimated from a batch of latent variable activations, and latent variables that fail to substantially decrease the fraction of false neighbors are de-weighted.

We summarize our regularizer here (see appendix for details): The regularizer $\mathcal{L}_{\text{FNN}}$ accepts as input a batch of $B$ coordinates of dimensionality $L$ corresponding to hidden activations $h \in \mathbb{R}^{B \times L}$. Here, $h$ corresponds to the network's current estimate of the attractor $\hat{Y}$ generated from an input comprising

$B$ length-$T$ rows randomly sampled from the full measurement matrix $X$. However, we note that our activity regularizer will work for any neural network with hidden layers, independently of the time series embedding problem studied here. Next, pairwise distances in latent space are computed among all points in the batch using only the first $m \leq L$ coordinates, resulting in a dimension-indexed Euclidean distance $D \in \mathbb{R}^{B \times B \times L}$. This matrix is sorted by column, and the first $K + 1$ columns are used to generate a list of $K$ batch indices corresponding to the $K$ nearest neighbors of each point $i$ when only the first $m \in \{1, 2, ..., L\}$ latent dimensions are considered. Array masking is used to vectorize this computation compared to loop-based implementations in earlier works [13]. The resulting nearest neighbor index matrix is then used to calculate the fraction of $K$-neighbors of each point $i$ that remain the same as the latent index $m$ increases. The fraction of surviving neighbors is then averaged across all points in the batch, producing a batch-averaged fraction of false neighbors $\bar{F} \in \mathbb{R}^L$ that arise at each latent index. The false-neighbors vector $\bar{F} \in \mathbb{R}^L$ then weights an activity regularizer $\mathcal{L}_{\text{FNN}} = \sum_{m=2}^{L}(1 - \bar{F}_m)\bar{h}_m^2$, where $\bar{h}_m$ is the batch-averaged activity of the $m^{th}$ latent unit . Altogether, the loss function for the autoencoder has the form

$$\mathcal{L}(X, \hat{X}, \hat{Y}) = \|X - \hat{X}\|^2 + \lambda \, \mathcal{L}_{\text{FNN}}(\hat{Y})$$

where $\|.\|^2$ denotes the Euclidean norm, and $\lambda$ is a hyperparameter controlling the relative strength of the regularizer.

## 4.2 Experiments

**Models.** We illustrate the utility of the loss function $\mathcal{L}_{\text{FNN}}$ across different architectures by using two standard encoder models for all experiments: a single-layer long short-term memory network (LSTM) and a three-layer multilayer perceptron (MLP), each with $L = 10$ latent units. These architectures were chosen because they have a comparable number of parameters (520 and 450, respectively), which is small compared to the minimum of 5000 timepoints used for all datasets (see supplementary material for additional model details). We obtain comparable results with both models, and we include the MLP results in the appendices. As baseline models, we use eigen-time-delay coordinates (ETD) [40, 42], time-lagged independent component analysis (tICA) [43], and unregularized replicates of the autoencoders ($\lambda = 0$). In the supplementary material, we also include baseline results for a previously-proposed autoencoder model, comprising a one-layer MLP with $\tanh$ activation [52].

Across all experiments, we only tune the regularizer strength $\lambda$ and the learning rate $\gamma$. Because blind embedding is an unsupervised learning problem, we do not change the network architecture, optimizer, and other hyperparameters. As a general heuristic, we adjust $\lambda$ to be just small enough to avoid dimensionality collapse in the reconstructed attractor (an easily-recognized phenomenon discussed in the next section), and we vary $\gamma$ only to ensure convergence within the constant number of training epochs used for all experiments. For all results, we train five replicate networks with random initializations.

**Datasets.** We study datasets corresponding to several chaotic or quasiperiodic systems: stochastic simulations of the three-dimensional Lorenz "butterfly" attractor, the three-dimensional Rössler attractor, a ten-dimensional Lotka-Volterra ecosystem, a three-dimensional quasiperiodic torus, and an experimental dataset corresponding to centroid measurements of a chaotic double pendulum (an effectively four-dimensional system over short timescales) [62]. For all datasets, 5000 timepoints are used to construct separate Hankel matrices for training and validation of $\mathbf{g}$ and $\mathbf{g}'$, and 5000 separate timepoints are used as a test dataset for embedding. For each dataset, different replicates or initial conditions are used for train and test partitions whenever possible; for single-series datasets, sets of 5000 timepoints separated by at least 1000 timepoints are excerpted to prevent overlap between train, validation, and test. For exploratory analysis of datasets with unknown governing equations, we use datasets corresponding to: temperature measurements of the irregularly-firing "Old Faithful" geyser; a human electrocardiogram; hourly electricity usage measurements for 321 households; and spiking rates for neurons in a mouse thalamus [63, 64, 65]. To ensure consistency across datasets, we downsample all time series to have matching dominant timescales (as measured by leading Fourier mode); otherwise, we apply no smoothing or detrending.

**Evaluation.** Because time series embedding constitutes an unsupervised learning problem, for testing performance against baselines, we train our models by choosing a single coordinate $y_1(t)$ from a known dynamical system $\mathbf{y}(t)$, which we use to construct a Hankel measurement matrix $X_{\text{train}}$. We then train our autoencoder on $X_{\text{train}}$, and then use it to embed the Hankel matrix of unseen data $X_{\text{test}}$

from the same system, producing the reconstruction $\hat{Y}_{\text{test}}$. We then compare $\hat{Y}_{\text{test}}$ to $Y_{\text{test}}$, a sample of the full attractor at the same timepoints. Because the number of latent coordinates $L$ is the same for all models, but the tested attractors have varying underlying dimensionality $d \leq L$, when comparing $Y$ to $\hat{Y}$ we lift the dimensionality of $Y$ by appending $L - d$ constant coordinates.

**Metrics.** We use several existing and novel methods to compute the similarity between the original attractor $Y$ and its reconstruction $\hat{Y}$. We emphasize that this comparison does not occur during training (the autoencoder only sees one coordinate); rather, we use these metrics to assess how well our unsupervised technique reconstructs known systems. We summarize these metrics here (see appendix for additional details):

*Pointwise comparison.* Before comparing $\hat{Y}$ and $Y$, we first apply the Procrustes transform, which applies translation, rotation, and reflection (but not shear) to align $\hat{Y}$ with $Y$. Because $X$ (and thus $\hat{Y}$) lacks information about the symmetry and chirality of $Y$, Procrustes alignment prevents relative orientation from affecting subsequent distance calculations. We then calculate the pointwise Euclidean distance, and normalize it to produce the Euclidean similarity. We obtain similar results using the dynamic time warping (DTW) distance, an alternative distance measure for time series [26]. Together, these metrics generalize previously-described metrics for comparing strange attractors [66].

*Forecasting.* We quantify the ability of the reconstructed attractor $\hat{Y}$ to predict future values of the original attractor $Y$, a key property of state space reconstructions used in causal inference [33]. We use the cross-mapping forecasting method [67]; in this algorithm, a simplex comprising the nearest neighbors of each point on $\hat{Y}$ are chosen, and then used to predict future values of each point on $Y$ at $\tau$ timesteps later. We average this forecast across all points, and then scale by the variance, to produce a similarity measure.

*Local neighborhoods.* We introduce a novel measure of global neighbor accuracy that describes the average number $\bar{\kappa}(k)$ of the $k$-nearest neighbors of each point on $\hat{Y}$ that also fall within the $k$-nearest-neighbors of the corresponding point on $Y$. This quantity is bounded between a perfect reconstruction, $\bar{\kappa}(k) = k$, and a random sort $\bar{\kappa}(k) = k^2/N$ (the mean of a hypergeometric distribution). Similar to an ROC-AUC, we compute similarity by summing $\bar{\kappa}(k)$ from $k = 1$ to $k = N - 1$, and scale its value between these two limits.

*Attractor dimensionality.* A central goal of our approach is determining an appropriate latent dimensionality $d_E$ for the attractor $\hat{Y}$. We use the variance of each latent coordinate as a continuous measure of its relative activity on the learned attractor, and we compare the variance per index between the embedding $\hat{Y}$ and full system $Y$. If the full system has $d < L$, we append $L - d$ constant (i.e. zero variance) dimensions to $Y$. We compute the mean square difference between the activity per index of the embedding and of the original system, and we scale this distance to generate a continuous measure of attractor dimension similarity, $\mathcal{S}_{\text{dim}}$.

*Topological features.* We quantify the degree to which $\hat{Y}$ retains essential structural features of $Y$, such as holes, voids, or the double scrolls of the Lorenz attractor. Following recent work showing that topological data analysis effectively captures global similarity between strange attractors [68, 37], we compute the Wasserstein distance between the persistence diagrams of $\hat{Y}$ and $Y$, which quantifies the presence of different topological features across length scales [69]. To produce a similarity measure, we normalize the this distance by the distance between the estimate $\hat{Y}$ and a null attractor with no salient features.

*Fractal dimension.* We calculate the similarity between the fractal dimensions of $\hat{Y}$ and $Y$. We use the correlation fractal dimension—rather than related quantities like Lyapunov exponents or Kolmogorov-Sinai entropy—because it can be calculated deterministically and non-parametrically from finite point sets [70].

## 5    Results

### 5.1    Reconstruction of known attractors

Figure 3A shows example embeddings of datasets with known attractors using the LSTM autoencoder with $\mathcal{L}_{\text{FNN}}$, illustrating the qualitative similarity of the learned embeddings to the true attractors.

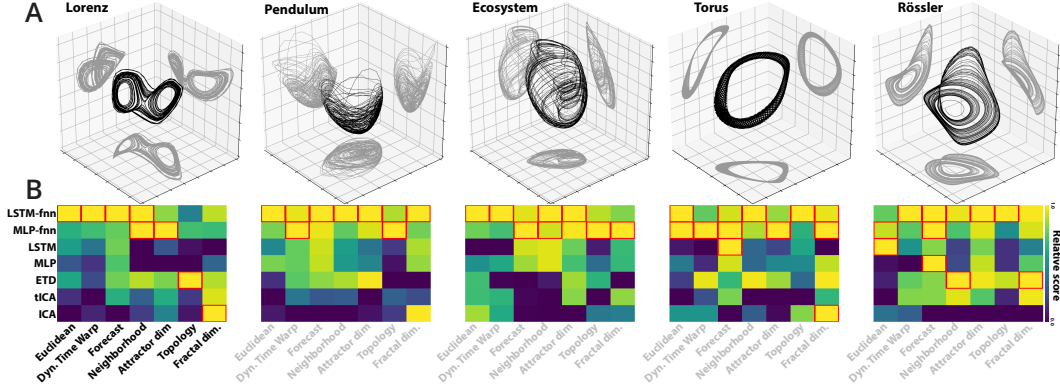

Figure 3: (A) Embeddings produced by the autoencoder with $\mathcal{L}_{\text{FNN}}$, trained on only the first coordinate of each system. (B) For each system, a variety of baseline embeddings are compared to the original attractor via multiple similarity measures. Hue indicates mean across 5 replicates scaled by column range, with red boxes indicating column maximum, or values falling within one standard deviation of it. Because distinct similarity metrics have different dynamic ranges, each column has been normalized separately to accentuate differences across models (see appendix for tabular values).

Figure 3B shows the results of extensive quantitative comparisons between the embeddings and the original attractors, across a variety of measures of attractor similarity (see appendix for raw values). Compared to baselines, the regularized network either matches or improves the quality of the embedding across a variety of different metrics and datasets. $\mathcal{S}_{\text{dim}}$ consistently improves with the regularizer, demonstrating that $\mathcal{L}_{\text{FNN}}$ fulfills its primary purpose of generating a latent space with appropriate effective dimensionality $d_E$. Importantly, this effect is not simply due to $\mathcal{L}_{\text{FNN}}$ indiscriminately compressing the latent space; for the ecosystem dataset, $d = L = 10$, and so achieving high $\mathcal{S}_{\text{dim}}$ requires that all latent units remain active after training. We also find that the qualitative appearance of our embeddings, as well as their effective dimensionality $d_E$ depends on $\lambda$ but not $L$ (see appendix). The other metrics encompass measures of cohomology, dynamical similarity, multivariate time series distance, and point cloud similarity, demonstrating that the learned embeddings improve on existing methods in several ways.

Importantly, we observe that the regularized autoencoder nearly always improves on the non-regularized model, suggesting that the regularizer has a clear and beneficial effect on the representations obtained by the model. We hypothesize that the stronger test performance of the regularized model occurs because the regularizer compresses the model more effectively than other latent regularization techniques such as lasso activity regularization (see appendix for comparison) thereby reducing overfitting without sacrificing dynamical information. We emphasize the consistency of our results across these datasets, which span from low-dimensional chaos (Lorenz and Rössler attractors), high-dimensional chaos (the ecosystem model), noisy non-stationary experimental data (the double pendulum experiment), and non-chaotic dynamics (the torus).

## 5.2 Forecasting noisy time series

Existing attractor reconstruction techniques are often sensitive to noise [19, 71]. This limitation may be fundamental: Takens' theorem and its corollaries provide no guarantee that a small perturbation to the attractor $Y$ will lead to a small perturbation to $\hat{Y}$. However, recent theoretical and numerical results have sought attractor reconstruction methods or measurement protocols that remain stable against noise [25, 15, 71]. We therefore quantify the robustness of our technique to noise by performing a series of simulations of the Lorenz equations that include time-dependent forcing by uncorrelated Brownian motion. We vary the relative amplitude of the the noise term, and then train separate models for each amplitude. We use the same hyperparameters as for the case without noise, as described above. Figure 4C shows the cross-mapping forecasting accuracy as a function of forecasting horizon, $\tau$, and the relative noise amplitude, $\xi_0 \in [0, 1]$ [67]. Consistent with the results for the attractor similarity measures, we find that the prediction accuracy decays the slowest for the regularized LSTM model, and that the advantage of the regularized model is more pronounced

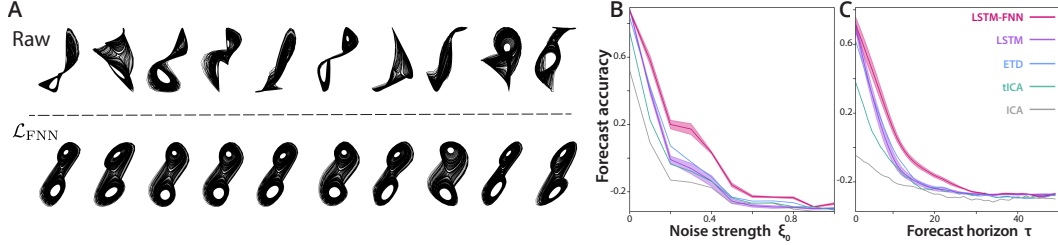

Figure 4: (A) Embeddings of the stochastic Lorenz dataset with and without the false-nearest-neighbors regularizer. Replicates correspond to different random initializations of the Brownian noise force and initial network weights. (B) The cross-mapping forecast accuracy as a function of noise strength $\xi_0$ (with constant $\tau = 20$). (C) The cross-mapping forecast accuracy versus forecasting horizon $\tau$ (with constant $\xi_0 = 0.5$). Standard errors span 5 replicates.

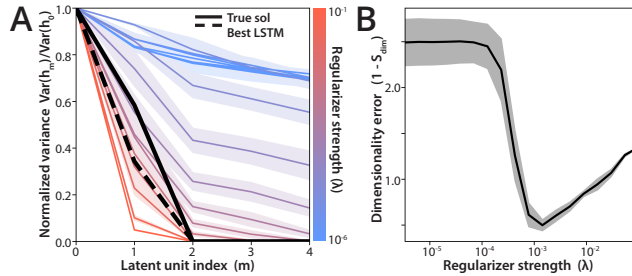

Figure 5: (A) The final distribution of latent variances for replicate networks trained on the Lorenz system, parametrized by regularizer strength; the normalized variance of the final best-performing LSTM (dashed black line) and the variance per coordinate of the full Lorenz system (solid black line) are overlaid. (B) The dimensionality error $1 - \mathcal{S}_{\text{dim}}$ versus $\lambda$. Standard errors span 5 replicates.

at long forecasting horizons. Moreover, when we train replicate networks with different random initializations (Figure 4A), we find that the regularized models consistently converge to similar sets of coordinates—suggesting that our method successfully identifies the salient signal in a noisy time series, and finds a general solution independent of the noise or initial weights.

## 5.3   Inferring the dimensionality of an attractor

We next investigate the effect of the regularizer strength $\lambda$ on the embedding. Figure 5A shows the effect of increasing the regularizer strength on the variance of the activations of the $L = 10$ ranked latent coordinates for embeddings of the Lorenz dataset. Identical experiments with the MLP model are included in the appendix. As $\lambda$ increases, the distribution of activation across latent variables develops increasing right skewness, eventually producing a distribution of activations similar to that of weighted principal components. Figure 5B shows the final dimensionality error $1 - \mathcal{S}_{\text{dim}}$ for replicate networks trained with different regularizer strengths. The plots show that the dimensionality accuracy of the learned representation improves as long as $\lambda$ is greater than a threshold value. However, the error begins to increase if $\lambda$ becomes too large, due to the learned attractor becoming overly flattened, and thus further from the correct dimensionality. This nonlinearity implies a simple heuristic for setting $\lambda$ for an unknown dataset: keep increasing lambda until the effective dimensionality of the latent space rapidly decreases, and then vary it no further.

## 5.4   Exploring datasets with unknown attractors

To demonstrate the potential utility of our approach for exploratory analysis of unknown time series, we next embed several time series datasets for which the governing equations are unknown, but for which low-dimensional attractors have previously been hypothesized. Figure 6 shows embeddings of various systems using the FNN loss with the LSTM model. For all systems, a different training dataset is used to construct $\mathbf{g}(.)$ than the testing dataset plotted. Several qualitative features of the embedded attractors are informative. For the electrocardiogram dataset, the model successfully creates a nested

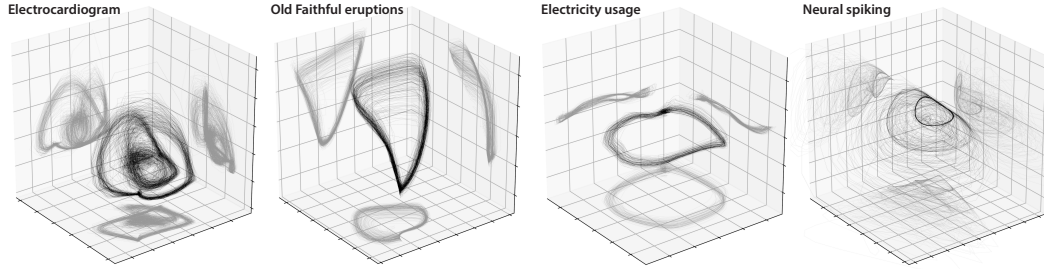

Figure 6: Embeddings of an electrocardiogram (160 heartbeats), temperature measurements of the erupting "Old Faithful" geyser in Yellowstone National Park (200 eruptions), average electricity usage by 321 households (200 days), and neural spiking in a mouse thalamus.

loop geometry reminiscent of that described in analytical models of the heart [72, 73]. This structure persists despite the plotted embedding corresponding to an ECG from a different patient than the one used to train the model. For the Old Faithful dataset, the model identifies a low-dimensional quasi-periodic attractor that is consistent with a long-speculated hypothesis that the geyser's nearly-regular dynamics arise from a strange attractor (spanning a small number of governing pressure and temperature state variables) [74]. The dense regions of the attractor correspond to eruption events—which occur with consistent, stereotyped dynamics—while the diffuse, fan-like region of the attractor corresponds to the slow recovery period between firings, which has a broader range of dynamics and timings. For the electricity usage dataset, the embedding reveals a circular limit cycle consistent with a stable daily usage cycle, in agreement with other time-series analysis algorithms [57]. For the mouse neuron spiking rate dataset, the model identifies a double-limit-cycle structure, consistent with higher-dimensional measurements suggesting that the neuronal dynamics lie on an intrinsic attractor manifold [65].

## 6 Discussion

We have introduced a method for reconstructing the attractor of a dynamical system, even when only low-dimensional time series measurements of the system are available. By comparing our technique to existing methods across a variety of complex time series, we have shown that our approach constructs informative, topologically-accurate embeddings that capture the intrinsic dimensionality of the original system. In practice, we are able to obtain strong results by tuning one hyperparameter, the regularizer strength $\lambda$, while adjusting the learning rate only to ensure convergence. Moreover, we empirically observe that the magnitude of $\lambda$ has a nonlinear effect on the structure of the embedding, providing a simple heuristic for tuning this hyperparameter in unsupervised settings. For this reason, we anticipate that our method can readily be applied to unknown time series datasets, which we have demonstrated using examples from areas spanning physiology, neuroscience, and geophysics. However, we note that our technique does introduce some tradeoffs compared to other methods of embedding time series: for simple or small datasets, lagged coordinates or eigen-time-delays are less computationally-expensive than our neural network based approach. However, for larger or more complex datasets, our method offers the same scaling advantages as other applications of neural networks to large datasets. Moreover, our general-purpose activity regularizer can be used with any network architecture and thus potentially other problem domains beyond time series embedding. Our technique and open-source implementation supports multidimensional time series, and in future work we hope to further draw upon classical results in the theory of chaotic systems in order to more directly relate quantitative properties of the learned attractors—such as the Lyapunov exponents and fractal dimension—to statistical features of the network's underlying representation of the time series. More broadly, we hope that our approach can inform efforts to learn differential equation models that describe latent dynamics [75, 3, 76], which have recently been shown to exhibit a tunable tradeoff between accuracy and parsimony [77]—an effect that may be mitigated by more constrained latent representations.

## Broader Impact

While our work is motivated by conceptual and theoretical problems in the theory of nonlinear and chaotic dynamical systems, our method incurs comparable ethical and societal impacts to other techniques for mining time series data. While we focus primarily on datasets from the natural sciences, our technique can be used to identify recurring patterns and motifs within industrial datasets, such as subtle usage patterns of a utility, or fluctuations in consumer demand. Likewise, our technique is well-suited to the analysis of data from fitness trackers, in which a small number of measured dynamical variables (acceleration, elevation, heart rate, etc) are used as a proxy for a model of an individual's behavior (and even overall health status). Both of these examples represent cases in which the technique may identify latent factors about individuals that they might not anticipate being observable to a third-party—thus introducing concerns about privacy. Avoiding such problems first requires intervention at the level of the data provided to the technique (e.g., aggregating time series across individuals before training the model) or strict retention limitation policies (e.g., providing an overall health score, and then deleting the underlying model behind that score). For example, in our technique, the rows of the Hankel matrix used to train the model can be randomly sampled from a pool of user time series, resulting in an aggregated model that avoids modeling any particular user.

However, the ability of our method to identify latent dynamical variables also motivates potential positive societal impacts, particularly in regards to potential benefits for the analysis of physiological data. We show examples of non-trivial structure being extracted from cardiac measurements and neural activity; these structures may be used to better identify and detect anomalous dynamics, potentially improving health monitoring. More broadly, better identification of latent factors within time series data may allow for more principled identification and removal of features that undermine individual privacy, although this would require *ex post facto* analysis and interpretation of latent variables discovered using our technique.

## Acknowledgments and Disclosure of Funding

We thank Chris Rycroft, Daniel Forger, Sigrid Keydana, Brian Matejek, Matthew Storm Bull, and Sharad Ramanathan for their comments on the manuscript. W. G. was supported by the NSF-Simons Center for Mathematical and Statistical Analysis of Biology at Harvard University, NSF Grant DMS 1764269, and the Harvard FAS Quantitative Biology Initiative. The author declares no competing interests.

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
