[Supplementary Material]



## CONTENTS

## I. SUPPLEMENTARY CODE

Code associated with this paper may be found at `https://github.com/williamgilpin/fnn`

## II. CALCULATION OF THE FALSE-NEAREST-NEIGHBOR REGULARIZER

Our loss function $\mathcal{L}_{\text{FNN}}$ represents a variational formulation of the false-nearest-neighbors method, a popular heuristic for determining the appropriate embedding dimension $d_E$ when using the method of lags [1]. The intuition behind the technique is that a $d$-dimensional embedding with too few dimensions will have many overlapping points, which will undergo large separation when the embedding is lifted to $d+1$. These points correspond to false neighbors, which only co-localize in $d$ dimensions due to having overlapping projections (Figure S1). The traditional false-nearest-neighbors technique asserts that the true embedding dimension $d_E$ occurs when the fraction of false nearest neighbors first approaches zero as $d$ increases.

Let $h \in \mathbb{R}^{B \times L}$ denote activations of a latent layer with $L$ units, generated when the network is given an input batch of size $B$. For the embedding problem studied here, $h$ corresponds to a partial embedding $\sim \hat{Y}$ generated from an input comprising $B$ length-$T$ rows randomly sampled from the full Hankel measurement matrix $X$. However, here we use general notation to emphasize that this regularizer can be applied to hidden layers in an arbitrary network.

We define the dimension-indexed, pairwise Euclidean distance $D \in \mathbb{R}^{B \times B \times L}$ among all points in the batch,

$$D^2_{abm} = \sum_{i=1}^{m} (h_{ai} - h_{bi})^2.$$

This tensor describes the Euclidean distance between samples $a$ and $b$ when only the first $m$ latent dimensions are considered. Calculation of this quantity therefore breaks ordering invariance among the latent dimensions.

We now define two related quantities: $\tilde{D}_{abm} \in \mathbb{R}^{B \times B \times L}$ corresponds to $D_{abm}$ sorted columnwise, while $\tilde{D}'_{abm} \in \mathbb{R}^{B \times B \times (L-1)}$ contains each column of $D_{abm}$ ordered by the sort order of the previous column. We calculate these quantities first by calculating the index tensor $g \in \mathbb{R}^{B \times B \times L}$, where each column $g_{a,:,m}$ contains the indices of all members of the batch sorted in ascending order of their relative distance from $a$ when only the first $m$ dimensions are considered. We then use $g$ to define

$$\tilde{D}_{abm} = \sum_{\beta=1}^{B} \delta_{\beta, g_{abm}} D_{a\beta m}, \quad \tilde{D}'_{abm} = \sum_{\beta=1}^{B} \delta_{\beta, g_{ab,m-1}} D_{a\beta m}.$$

These quantities allow computation of the normalized change in distance to a given neighbor as $m$ increases,

labelled by its relative distance, $S_{abm} = (\tilde{D}'^2_{abm} - \tilde{D}^2_{abm})/\tilde{D}^2_{abm}$, where $m \geq 2$.

A *false neighbor* is an $m-1$ dimensional near-neighbor that undergoes a jump greater than $R_{\text{tol}}$ when lifted to $m$ dimensions. We therefore define a binary tensor describing whether each point $a$ undergoes a jump of this magnitude in its $m^{th}$ dimension,

$$R_{abm} = \begin{cases} 1 & S_{abm} \geq R_{\text{tol}} \\ 0 & S_{abm} < R_{\text{tol}} \end{cases}.$$

The threshold $R_{\text{tol}}$ can be chosen arbitrarily; in practice we find that it has little effect on our results, and so we set it to a constant value $R_{\text{tol}} = 10$ (a standard value) for all experiments [1].

In regions of the attractor where the dynamics proceeds relatively quickly, the uniformly-spaced time series comprising $\hat{Y}$ undersamples the attractor. This can lead to points undergoing large shifts in position relative to the scale of the attractor as $m$ increases, leading to an additional criterion for whether a given point is considered a false neighbor. We define the characteristic size of the attractor in the first $m$ latent coordinates,

$$\mathcal{R}_m^2 = \frac{1}{m\,B} \sum_{b=1}^{B} \sum_{i=1}^{m} (h_{bi} - \bar{h}_i)^2,$$

where $\bar{h}_i = (1/B) \sum_{b=1}^{B} h_{bi}$. This quantity defines a second criterion,

$$A_{akm} = \begin{cases} 1 & \tilde{D}_{abm} \geq A_{\text{tol}}\mathcal{R}_m \\ 0 & \tilde{D}_{abm} < A_{\text{tol}}\mathcal{R}_m \end{cases}.$$

The behavior of the regularizer does not strongly vary with $A_{\text{tol}}$, as long as this hyperparameter is set to a sufficiently large value. We therefore set $A_{\text{tol}} = 2.0$, a standard value in the literature, and keep it constant for all experiments.

We define the elementwise false neighbor matrix, which indicates points that satisfy either or both of these criteria,

$$F_{abm} = \Theta(R_{abm} + A_{abm})$$

where $\Theta$ denotes the left-continuous Heaviside step function, $\Theta(x) = 1, x > 0$, $\Theta(x) = 0, x \leq 0$. We next contract dimensionality by averaging this quantity $F_{abm}$ across both the batch and the set of $K$ nearest neighbors to $a$,

$$\bar{F}_m = \frac{1}{K\,B} \sum_{k=1}^{K} \sum_{b=1}^{B} F_{kbm}.$$

The hyperparameter $K$ determines how many neighbors are considered close enough to be informative about the topology of the attractor. Because varying this hyperparameter has a similar effect to changing $B$, we set $K = \max(1, \lceil 0.01B \rceil)$ and otherwise leave this parameter constant; as with the original false-nearest-neighbors

Figure S1. A set of near neighbors in a two-dimensional projection of three-dimensional point cloud (circled blue and red points). False neighbors (red) separate when the system is lifted to a higher dimension.

method, our approach performs well even when $K = 1$ [1]. Having obtained the dimension-wise fractional false neighbor count $\bar{F}_m$, we now calculate the false neighbor loss,

$$\mathcal{L}_{\text{FNN}} = \sum_{m=2}^{L} (1 - \bar{F}_m)\bar{h}_m^2.$$

where $\bar{F}_m, \bar{h}_m$ and thus $\mathcal{L}_{\text{FNN}}$ implicitly depend on the batch activations $h$. Overall, $\mathcal{L}_{\text{FNN}}$ has the form of an activity regularizer acting on the latent coordinates. The overall loss function for the autoencoder is therefore

$$\mathcal{L}(X, \hat{X}, \hat{Y}) = \|X - \hat{X}\|^2 + \lambda\,\mathcal{L}_{\text{FNN}}(\hat{Y})$$

where $\|.\|^2$ denotes the mean square error averaged across the batch, and $\lambda$ is a hyperparameter controlling the relative strength of the regularizer.

## III. DESCRIPTION OF REFERENCE DATASETS

**Lorenz attractor.** The Lorenz equations are given by

$$\dot{x} = \sigma(y - x) \tag{A1}$$
$$\dot{y} = x(\rho - z) - y \tag{A2}$$
$$\dot{z} = xy - \beta z \tag{A3}$$

We use parameter values $\sigma = 10$, $\rho = 28$, $\beta = 2.667$. The system is simulated for 500 timesteps, with a stepsize $\Delta t = 0.004$. The system is then downsampled by a factor of 10. We fit the model using $x(t)$, which we divide into separate train, validation, and test datasets comprising 5000 timepoints sampled from three trajectories with different initial conditions. To avoid transients, for each partition we select the last 5000 timepoints from a 125000 step trajectory. For stochastic simulations of this system, an uncorrelated white noise term $\xi(t)$, $\langle \xi(t)\xi(t') \rangle = \xi_0^2 \delta(t - t')$ is appended to each dynamical variable before integration,

the integration timestep is decreased to $\Delta t = 0.0004$, and the integration output is downsampled by a factor of 100.

**Rössler attractor.** The Rössler attractor is given by

$$\dot{x} = -y - z \tag{A4}$$
$$\dot{y} = x + ay \tag{A5}$$
$$\dot{z} = b + z(x - c) \tag{A6}$$

We use parameter values $a = 0.2$, $b = 0.2$, $c = 5.7$, which produces a chaotic attractor with the shape of a Möbius strip. The system is simulated for 2500 timesteps, with a stepsize $\Delta t = 0.125$. The system is then downsampled by a factor of 10. We fit the model using $x(t)$, which we divide into separate train, validation, and test datasets corresponding to different initial conditions.

**Ecological resource competition model.** We use a standard resource competition model, a variant of the Lotka-Volterra model that is commonly used to describe scenarios in which $n$ distinct species compete for a pool of $k$ distinct nutrients. We let $N_i(t)$ denote the abundance of species $i$, and $R_j(t)$ denote the availability of resource $j$.

$$\dot{N}_i = N_i\left(\mu_i(R_1, ..., R_k) - m_i\right) \tag{A7}$$

$$\dot{R}_j = D(S_j - R_j) - \sum_{i=1}^{n} c_{ji}\,\mu_i(R_1, ..., R_k)N_i \tag{A8}$$

where the species-specific growth rate is given by

$$\mu_i(R_1, ..., R_k) = \min\left(\frac{r_i R_1}{K_{1i} + R_1}, ..., \frac{r_i R_k}{K_{ki} + R_k}\right).$$

This model is strongly chaotic for a range of parameter values, and it was recently used to argue that chaotic dynamics may account for the surprising stability in long-term population abundances of competing phytoplankton species in the ocean [2]. We use parameter values from this study, which corresponds to $n = 5$ species and $k = 5$ resources. The full parameter values are: $D = 0.25$, $r_i = r = 1$, $m_i = m = 0.25$, $\mathbf{S} = [6, 10, 14, 4, 9]$, $\mathbf{K} =$

$$\begin{bmatrix} 0.39 & 0.34 & 0.3 & 0.24 & 0.23 \\ 0.22 & 0.39 & 0.34 & 0.3 & 0.27 \\ 0.27 & 0.22 & 0.39 & 0.34 & 0.3 \\ 0.3 & 0.24 & 0.22 & 0.39 & 0.34 \\ 0.34 & 0.3 & 0.22 & 0.2 & 0.39 \end{bmatrix},$$

$$\mathbf{c} = \begin{bmatrix} 0.04 & 0.04 & 0.07 & 0.04 & 0.04 \\ 0.08 & 0.08 & 0.08 & 0.1 & 0.08 \\ 0.1 & 0.1 & 0.1 & 0.1 & 0.14 \\ 0.05 & 0.03 & 0.03 & 0.03 & 0.03 \\ 0.07 & 0.09 & 0.07 & 0.07 & 0.07 \end{bmatrix}.$$

We simulate this system for 200000 units of time, with timestep $\Delta t = 0.1$. We discard the first 100000 timepoints to eliminate any transients, and then downsample the time series by a factor of 10. We fit the model using $R_1(t)$, which we divide into separate train, validation, and test datasets corresponding to different initial conditions.

**Three-dimensional torus.** We parametrize a torus as a continuous-time, quasiperiodic dynamical system

$$\dot{x} = -an\sin(nt)cos(t) - (r + a\cos(nt))\sin(t) \tag{A9}$$
$$\dot{y} = -an\sin(nt)\sin(t) + (r + a\cos(nt))\cos(t) \tag{A10}$$
$$\dot{z} = an\cos(nt) \tag{A11}$$

where we use the parameters $r = 1$ (the outer radius), $a = 1/2$ (the cross-sectional radius), $n = 15.3$ (the winding number). Because $n$ is not an integer, trajectories of this system are non-recurring and quasiperiodic. The system is simulated for 2000 timesteps, with a stepsize $\Delta t = 0.02$. The time series is then downsampled by a factor of 8. We fit the model using $x(t)$, which we divide into separate train, validation, and test datasets corresponding to different initial conditions.

**Double pendulum experimental dataset.** We use an existing experimental dataset comprising a 400 fps video of a double pendulum experiment, recorded on a high-speed Phantom Miro EX2 camera [3]. The video was segmented by the original authors, and the centroid positions were recorded for the pivot attachment to the wall, the joint between the first and second pendula, and the tip of the second pendulum. We convert this dataset into new time series corresponding to the angles that the first and second pendulum make with the vertical direction, $(\theta_1, \theta_2)$. These time series are then numerically differentiated, in order to produce a time series of the angular velocities $(\dot{\theta}_1, \dot{\theta}_2)$. For an ideal double pendulum, the four coordinates $(\dot{\theta}_1, \dot{\theta}_2, \theta_1, \theta_2)$ canonically parametrize the Hamiltonian of the system, and so these four coordinates are used as the definition of the attractor. However, we note that, for the experimental dataset, the time-averaged kinetic energy $T \propto \dot{\theta}_1^2 + \dot{\theta}_2^2$ gradually decreases throughout the course of the experiment. This additional coordinate was not included in the reference description of the attractor, due to its slow dynamics and non-stationarity, and so it constitutes an external, non-autonomous source of variation for which the model must compensate.

We downsample the raw time series by a factor of 3 and us $\dot{\theta}_1(t)$ as the input to the model. For training and validation, we use the first and second sequences of 5000 timepoints from the first experimental dataset. For testing, we use the first 5000 timepoints from the second experimental dataset.

## IV. DESCRIPTION OF EXPLORATORY DATASETS

**Electrocardiogram.** We use recordings from the PhysioNet QT database, which comprises fifteen-minute, two-lead ECG recordings from multiple individuals [4, 5]. Measurements are spaced 0.004 seconds apart. To remove high-frequency components, datasets were smoothed with a third-order Savitzky-Golay filter with a window size of 15 timepoints. The datasets are then downsampled by

a factor of 10. For the analysis presented here, we use 10000 datapoints (post-subsampling) from the dataset `sel102.dat` as training data, and for testing data we use 10000 datapoints from the dataset `sel103.dat` (which corresponds to a different patient).

**Electricity usage.** We use a dataset from the UCI machine learning database [6, 7], comprising residential power consumption by 321 Portuguese households between 2012 and 2014. Raw data is measured in units of kilowatts times the fifteen minute sampling increment. We create a consolidated dataset by taking the mean of all residences at each timepoint, adjusting the sample size as necessary at each timepoint to account for missing values for some households. We use the first, second, and last 10000 timepoints training, validation, and testing data.

**Geyser temperature measurements.** We use temperature recordings from the GeyserTimes database (`https://geysertimes.org/`), which consist of temperature readings from the main runoff pool of the Old Faithful geyser, located in Yellowstone National Park. Temperature measurements start on April 13, 2015 and occur in one-minute increments. The dataset was detrended by subtracting out a version of the data smoothed with a moving average over a one-day window, which effectively removes gradual effects like seasonal variation from the attractor. For the analysis presented in the main text, we use the first, second, and last 10000 datapoints from the Old Faithful dataset as training, validation, and test datasets, respectively, corresponding to $\approx 400$ eruptions of the geyser.

**Neural spiking.** We use a dataset from a recent study characterizing the intrinsic attractor manifold of neuron firings in freely-moving mice [8]. The raw spike count data is available from the CRCNS database (`http://crcns.org/data-sets/thalamus/th-1`), and we process this data using the authors' included code and instructions, in order to generate time series corresponding to spiking rates for single neurons. We use the first, second, and last 10000 timepoints training, validation, and testing data.

## V. MODELS

We apply eigen-time-delay (ETD) embedding as in previous studies [9], using principal component analysis as implemented in `scikit-learn` [10]. We apply time-structure independent component analysis (tICA) as implemented in the `MSMBuilder` software suite [11]. For numerical integration of chaotic systems, we use the LSODA method as implemented in `scipy` [12].

Autoencoders are implemented using TensorFlow [13]. The LSTM autoencoder has architecture: [Input-GN-LSTM(10)-BN]-[GN-LSTM(10)-BN-ELU-Output]. The multilayer perceptron has architecture: [Input-GN-FC(10)-BN-ELU-FC(10)-BN-ELU-FC(10)-BN]–[GN-FC(10)-BN-ELU-FC(10)-BN-ELU-FC(10)-BN-ELU-Output]. ELU denotes an exponential linear unit with default scale parameter 1.0, BN denotes a BatchNorm layer, GN denotes a Gaussian noise regularization layer (active only during training) with default standard deviation 0.5, and $FC$ denotes a fully-connected layer. 10 hidden units are used in all cells, including for the latent space $L = 10$, and network architecture or structural hyperparameters are kept the same across experiments. For both architectures, no activation is applied to the layer just before the latent layer, because the shape of the activation function is observed to constrain the range of values in latent space, consistent with prior studies [14].

## VI. EXTENDED DESCRIPTION OF SIMILARITY METRICS

**Evaluation metrics.** We introduce several methods for comparing the original system $Y$ with its reconstruction $\hat{Y}$. We emphasize that this comparison does not occur during training (the autoencoder only sees one coordinate); rather, we use these metrics to assess how well our models can reconstruct known systems.

*1. Dimension accuracy.* A basic, informative property of a dynamical system $\dot{\mathbf{y}}(t)$ is its dimensionality, $d = \dim(\mathbf{y})$, the minimum number of distinct variables necessary to fully specify the dynamics. Embeddings with $d_E < d$ discard essential information by collapsing independent coordinates, while embeddings with $d_E > d$ contain redundancy. We thus introduce a measure of embedding parsimony based on the effective number of latent coordinates present in the learned embedding.

We equate the activity of a given latent dimension with its dimension-wise variance $\text{Var}(\hat{\mathbf{y}})$, calculated across the ensemble of model inputs $\{\mathbf{x}_i\}_1^N$. We compare the distribution of activity in the reconstruction $\hat{Y}$ to the original attractor $Y$, padding the dimensionality of the original attractor with zeros as needed:

$$\mathcal{S}_{dim} = 1 - \frac{||\text{SORT}(\text{Var}(\mathbf{y})) - \text{SORT}(\text{Var}(\hat{\mathbf{y}}))||}{||\text{Var}(\mathbf{y})||}. \quad \text{(A12)}$$

This quantity is maximized when the number of active latent dimensions, and their relative activity, matches that found in the original attractor. We further discuss this score, and general properties of the embedding dimension $d_E$, in the next section.

*2. Procrustes distance.* Because a univariate measurement cannot contain information about the symmetry group or chirality of the full attractor, when computing pointwise similarity between the true and embedded attractors, we first align the two datasets using the Procrustes transform,

$$P = \arg\min_{\tilde{P}} \|\tilde{P}\hat{Y} - Y\|_F \quad \text{s.t.} \quad \tilde{P}^\top \tilde{P} = I,$$

where $I$ is the identity matrix. This transformation linearly registers the embedded attractor to the original

attractor via translation, rotation, reflection, but *not* shear. For example, after this transformation, mirror images of a spiral would become congruent, whereas a sphere and ellipsoid would not. After calculating this transform, we compute the standard Euclidean distance, which we normalize to produce a similarity metric,

$$\mathcal{S}_{\text{proc}} = 1 - \frac{||P\hat{Y} - Y||}{||Y - \bar{Y}||}$$

where the mean square error $||.||^2$ is averaged across the batch, and $\bar{Y}_k = \sum_{b=1}^{N} Y_{kb}$. This metric corresponds to a weighted variant of a classical attractor similarity measure [15]. In addition to the mean-squared error, we also calculate the dynamic time warping (DTW) distance between $P\hat{Y}$ and $Y$, which yields similar results as $\mathcal{S}_{\text{proc}}$.

*3. Persistent Homology.* The persistence diagram for a point cloud measures the appearance or disappearance of essential topological features as a function of length scale. A length scale $\epsilon$ is fixed, and then all points are replaced by $\epsilon$-radius balls, the union of which defines a surface. Key topological features (e.g., holes, voids, and extrema) are then measured, the parameter $\epsilon$ is increased, and the process is repeated. This process produces a birth-death diagram for topological features parametrized by different length scales. We refer to a recent review [16] for further details of the technique. Here, we build upon recent results showing that the Wasserstein distance between two persistence diagrams can be used as a measure of topological similarity between two dynamical attractors [17, 18]. We express this quantity as a normalized similarity measure

$$\mathcal{S}_{\text{homol}}(\mathcal{P}_Y, \mathcal{P}_{\hat{Y}}) = 1 - \frac{d_b(\mathcal{P}_Y, \mathcal{P}_{\hat{Y}})}{d_b(\mathcal{P}_Y, 0)}$$

where $\mathcal{P}_Y, \mathcal{P}_{\hat{Y}}$ denote the persistence diagrams associated with the point clouds $Y$ and $\hat{Y}$, and the denominator denotes distance to a "null" diagram with no salient topological features. Two attractors will have a high Wasserstein similarity if they share essential topological features (such as holes, voids, and extrema). We compute birth-death persistence diagrams using the `Ripser` software package [19], and we compute Wasserstein distances between diagrams using the `persim` software package [20].

*4. Local neighbor accuracy.* We seek to quantify whether points on $\hat{Y}$ are embedded in the same neighborhood as they are on $Y$, using simplex cross-mapping [21, 22]. We summarize this technique here: We pick a single datapoint $\hat{\mathbf{y}}_i$ from the attractor $\hat{Y}$, and then find the set $\{j\}_1^k$ comprising its $k$ nearest neighbors on $\hat{Y}$. Following standard practice, we use the minimum number of neighbors to form a bounding simplex, $k = d_E + 1$ [21]. We then select the corresponding $\{j\}_1^k$ points from the attractor $Y$, producing the set $\{\mathbf{y}_j\}_1^k$. The centroid of $\{\mathbf{y}_j\}_1^k$ is used to generate an estimate $\tilde{\mathbf{y}}_\mathbf{j}$ for the position of point $\mathbf{y}_j$. The procedure is repeated for all values of $i$, and the difference between $\tilde{\mathbf{y}}_\mathbf{j}$ and $\mathbf{y}_j$ averaged across all points is used as the distance measure

between $\hat{Y}$ and $Y$. In order to generate a time-delayed prediction, a factor $\tau$ is added to the indices of all points in $\{j\}_1^k$. Following previous work, we convert this distance into a similarity metric $\mathcal{S}_{\text{simp}}$ by normalizing by the dimensionwise-summed variance of the positions of all points in $Y$, and then subtracting the resulting quantity from one [23]. Generally $\mathcal{S}_{\text{simp}}$ decreases smoothly with $\tau$, and so we report results for several values of $\tau$.

*5. Global neighbor coverage.* For the $i^{th}$ point of the $N$ embedded points in $\hat{Y}$, we define $\kappa_i(k)$ as the number $k$ nearest neighbors that correspond to true neighbors in the original dataset $Y$. For example, if the indices of the three closest neighbors to point 1 in $Y$ are $11, 14, 29$ in order of relative distance, whereas its three closest neighbors are $11, 29, 15$ in $\hat{Y}$, then $\kappa_1(1) = 1, \kappa_1(2) = 1, \kappa_1(3) = 2$. We average this quantity across all points in $\hat{Y}$, $\bar{\kappa}(k) = \sum_{b=1}^{B} \kappa_b(k)$. We note that, for a random shuffling of neighbors, $\kappa(k)$ is given by the hypergeometric distribution describing a random sample of $k$ objects from a collection of $N$ distinct objects without replacement, $\kappa(k) \sim f(N, N, k), \bar{\kappa}(k) = k^2/N$; in contrast, a set of perfectly matching neighbors will exhibit $\bar{\kappa}(k) = k$. We use these bounds to define the neighbor similarity as the area under the curve between the observed $\bar{\kappa}(k)$ and the random case, normalized by the best-case-scenario

$$\mathcal{S}_{\text{nn}} = \frac{1}{N} \sum_{k=1}^{N-1} \frac{\bar{\kappa}(k) - k^2/N}{k - k^2/N}$$

Similar to an ROC AUC, this metric depends on the fraction of correct neighbors within the closest $k$ neighbors, as the parameter $k$ is swept. We illustrate calculation of this quantity diagrammatically in Figure S2.

*6. Fractal dimension.* As an example of a physically-informative quantity that can be computed for an attractor, but not a raw time series, we compare the correlation dimension (a type of fractal dimension) of the original attractor $c_Y$ and its reconstruction $c_{\hat{Y}}$ using the symmetric mean absolute percent error

$$\mathcal{S}_{\text{corr}}(c_Y, c_{\hat{Y}}) = 1 - \frac{|c_Y - c_{\hat{Y}}|}{|c_Y| - |c_{\hat{Y}}|}.$$

We use the correlation dimension instead of related physical properties (such as the Lyapunov exponent, or Kolmogorov-Sinai entropy) because, unlike other properties, the correlation dimension can be robustly measured in a parameter-free manner, without random subsampling of points [24].

## VII. ADDITIONAL EXPERIMENTS

### A. Application to time series clustering

Outside of physics, a significant application of attractor reconstruction lies in improving the representation and featurization of time series datasets [6, 25, 26]. We apply

Figure S2. Calculation of the nearest-neighbor coverage metric, $\mathcal{S}_{nn}$. (Left) the number of matching $k$ nearest neighbors as a function of $k$ for two identical point clouds, an empirical reconstruction of a point cloud, and a cloud of random points (for which the fraction of matching nearest neighbors is given by the hypergeometric distribution). (Right) the cumulative sum of the quantities on the left, scaled to lie in the interval between the two values.

our technique to four time series classification tasks from different application domains: (1) a synthetic dataset consisting of the $x$ coordinate of simulations of the Lorenz equations with different initial conditions, labelled by the exact values of the parameters defining the equations, (2) the first principle component of the body shape of crawling *C. elegans* worms, labelled by the genetic mutant; (3) electrocardiogram recordings of patients undergoing either standing, walking, or jumping; (4) electroencephalogram measurements of patients imagining one of two possible movements [5, 27, 28]. We do not tune hyperparameters, and instead use the same default hyperparameters used to train the Lorenz attractor in the previous experiments. We use a 1-nearest-neighbor classifier with dynamic time warping (a standard baseline for time series classification) [29], and summarize our results in Table S1. Across a variety of data sources and numbers of classes, classifiers using attractors obtained from our method achieve higher balanced accuracy than classifiers trained on the bare time series, or that use alternative embedding techniques. We obtain these results with no hyperparameter tuning, demonstrating that our method can generically extract meaningful features at each point in a time series—suggesting potential application of our approach as an initial featurization stage for general time series analysis techniques.

### B. Consistency and repeatability

We evaluate the repeatability and consistency of the learned representations by training an ensemble of models on the Lorenz dataset. All hyperparameters are held constant, and the only difference across replicates is the random weight initialization. As a baseline, we also trained

Table S1. The balanced classification accuracy for different time series. The number of classes in each dataset is indicated in parentheses.

| Dataset | Raw | tICA | ETD | LSTM | LSTM-fnn |
|---|---|---|---|---|---|
| Lorenz (8) | 0.18 | 0.22 | 0.18 | 0.21 | **0.23** |
| Worm (5) | 0.52 | 0.45 | 0.39 | 0.60 | **0.61** |
| ECG (3) | 0.40 | 0.20 | **0.47** | 0.40 | **0.47** |
| EEG(3) | 0.46 | 0.43 | 0.43 | 0.44 | **0.51** |

a set of models with no false-neighbors regularization. Example embeddings of the test data for models with and without regularization are shown in Figure S3. Before plotting, the Procrustes transform was used to remove random rotations.

The figure demonstrates the regularizer produces significantly more consistent embeddings across replications, implying that the regularizer successfully constrains the space of latent representations. We quantify this effect by computing the pairwise topological similarity $\mathcal{S}_{homol}$ among all replicates (Table S2), and we observe that the median topological similarity is larger for the regularized models.

Figure S3. An ensemble of reconstructed attractors for the Lorenz dataset, generated by models with different initial random weight initializations but identical hyperparameters. Upper portion of the plot shows models with no regularizer, and lower portion shows models with the false-nearest-neighbors regularizer. Before plotting, attractors were aligned using the Procrustes transform in order to remove random rotations.

### C. Effect of regularizer on alternate models

We repeat the experiment (described in the main text) in which the regularizer strength $\lambda$ is varied, and show sim-

Table S2. The median and standard error of the median across 20 replicate models.

| | LSTM | LSTM-FNN |
|---|---|---|
| $\langle \mathcal{S}_{\text{HOMOL}} \rangle$ | $0.09 \pm 0.05$ | $0.21 \pm 0.07$ |

ilar results for both the LSTM and the MLP autoencoders in Figure S6.

Figure S5. The final dimensionality error versus regularizer strength for the Lorenz dataset (A) and the double pendulum dataset (B). Error ranges correspond to 5 replicates.

### E. Comparison to a vanilla activity regularizer

We also compare the false-neighbor regularizer to a standard L1 activity regularizer (across a variety of different regularizer strengths), and find that the false-neighbor regularizer shows improvement across the different metrics used in the main text.

Figure S4. (A) The distribution of normalized latent variances as a function of regularizer strength from $\lambda = 0$ (blue) to $\lambda = 0.1$ (red), with the normalized variance for the full solution (solid black line) and for the final best-performing model (dashed black line). (B) The dimensionality error $1 - \mathcal{S}_{\text{dim}}$ as a function of $\lambda$. Error ranges correspond to 5 replicates.

### D. Dimension error versus regularizer strength for pendulum dataset

In Figure S5, we repeat the experiment (described in the main text) in which we vary the regularizer strength ($\lambda$), and we observe that the final dimensionality error $\mathcal{S}_{\text{dim}}$ exhibits a similar nonlinear dependence on $\lambda$ for both the double pendulum and Lorenz datasets.

Figure S6. Reconstruction accuracies for an LSTM model on the Lorenz dataset, using the false-nearest-neighbors regularizer (red) and a standard L1 activity regularizer (blue) on the latent units. While the regularizer strength is varied, all other hyperparameters are held constant at the values used in other experiments.

## F. Dependence on the number of latent units

In order to determine whether the hyperparameter $L$ influences the learned representations, we trained replicate autoencoders with the false-nearest-neighbor regularizer for $L = 10, 20, 30, 40, 50$ latent units (Figure S7). For all experiments, we used the Lorenz dataset with the same hyperparameters as were used in the main text. The figure shows that the relative activity of each latent unit after training is independent of $L$, implying that the network successfully learns to ignore excess latent dimensionality. Therefore, we argue that autoencoders trained with the false-nearest-neighbor regularizer will learn consistent representations that are determined primarily by the dataset, and by the relative strength of the regularizer $\lambda$.

Figure S8. (A) The cross-mapping forecast accuracy as a function of noise strength $\xi_0$ (with constant $\tau = 20$). (B) The cross-mapping forecast accuracy versus forecasting horizon $\tau$ (with constant $\xi_0 = 0.5$). Ranges correspond to standard error across 5 random initializations.

Figure S7. (A) Activity patterns of the first 10 latent units for autoencoders trained with $L = 10, 20, 30, 40, 50$ units in the latent space (blue to red). (B) The dimension accuracy score vs $L$

## VIII. ALL ATTRACTOR COMPARISON RESULTS

Table S3 shows the full results of attractor comparison experiments with all datasets, models, and metrics.

## G. Forecasting comparison to previous autoencoders

Previous work has proposed using a one-layer autoencoder with tanh activation in order to embed strange attractors, with the reconstruction loss serving as the sole loss function [30]. This prior work primarily focused on a forecasting task similar to our "noisy forecasting" experiments—the main difference being that we study simulations of stochastic differential equations (i.e. deterministic systems with added stochastic forcing), while the earlier work focuses on noisy measurements of a deterministic system. We re-implemented this earlier architecture and loss function, and compared it to our experiments with the stochastic Lorenz dataset (Figure S8, yellow traces). We find that the prior model performs slightly worse than our baseline unregularized LSTM, further underscoring the importance of the false-nearest-neighbors regularizer in generating consistent, predictive representations.

Table S3. Results for five datasets with known attractors. Errors correspond to standard errors over 5 replicates with random initial weights.

| METRIC | ICA | tICA | ETD | MLP | LSTM | MLP-FNN | LSTM-FNN |
|---|---|---|---|---|---|---|---|
| **LORENZ** | | | | | | | |
| $\mathcal{S}_{\text{SIMP}}$ | 0.42 | 0.74 | 0.82 | $0.79 \pm 0.03$ | $0.82 \pm 0.02$ | $0.81 \pm 0.03$ | $0.93 \pm 0.02$ |
| $\mathcal{S}_{\text{CORR}}$ | 0.992 | 0.985 | 0.978 | $0.91 \pm 0.01$ | $0.87 \pm 0.02$ | $0.953 \pm 0.009$ | $0.98 \pm 0.02$ |
| $\mathcal{S}_{\text{HOMOL}}$ | 0.049 | 0.123 | 0.668 | $0.01 \pm 0.01$ | $0.04 \pm 0.03$ | $0.47 \pm 0.05$ | $0.3 \pm 0.1$ |
| $\mathcal{S}_{\text{PROC}}$ | -0.015 | 0.037 | 0.212 | $0.09 \pm 0.05$ | $0.20 \pm 0.03$ | $0.23 \pm 0.08$ | $0.37 \pm 0.02$ |
| $\mathcal{S}_{\text{DTW}}$ | 0.237 | 0.21 | 0.27 | $0.25 \pm 0.04$ | $0.31 \pm 0.03$ | $0.39 \pm 0.09$ | $0.47 \pm 0.03$ |
| $\mathcal{S}_{\text{NN}}$ | 0.277 | 0.296 | 0.384 | $0.25 \pm 0.06$ | $0.25 \pm 0.06$ | $0.40 \pm 0.02$ | $0.40 \pm 0.02$ |
| $\mathcal{S}_{\text{DIM}}$ | 0.171 | 0.394 | 0.628 | $-0.4 \pm 0.1$ | $-0.06 \pm 0.08$ | $0.88 \pm 0.02$ | $0.66 \pm 0.01$ |
| **DOUBLE PENDULUM** | | | | | | | |
| $\mathcal{S}_{\text{SIMP}}$ | -0.24 | -0.06 | 0.30 | $0.35 \pm 0.03$ | $0.36 \pm 0.02$ | $0.39 \pm 0.02$ | $0.41 \pm 0.01$ |
| $\mathcal{S}_{\text{CORR}}$ | 0.985 | 0.861 | 0.822 | $0.96 \pm 0.01$ | $0.966 \pm 0.006$ | $0.951 \pm 0.003$ | $0.986 \pm 0.008$ |
| $\mathcal{S}_{\text{HOMOL}}$ | 0.191 | 0.202 | 0.176 | $0.18 \pm 0.03$ | $0.19 \pm 0.02$ | $0.26 \pm 0.03$ | $0.25 \pm 0.02$ |
| $\mathcal{S}_{\text{PROC}}$ | 0.002 | 0.001 | 0.003 | $0.013 \pm 0.004$ | $0.008 \pm 0.005$ | $0.013 \pm 0.003$ | $0.016 \pm 0.006$ |
| $\mathcal{S}_{\text{DTW}}$ | 0.026 | 0.069 | 0.108 | $0.11 \pm 0.02$ | $0.11 \pm 0.01$ | $0.136 \pm 0.009$ | $0.132 \pm 0.008$ |
| $\mathcal{S}_{\text{NN}}$ | 0.019 | 0.031 | 0.055 | $0.041 \pm 0.003$ | $0.042 \pm 0.002$ | $0.05 \pm 0.001$ | $0.060 \pm 0.001$ |
| $\mathcal{S}_{\text{DIM}}$ | -1.772 | -1.914 | 0.927 | $-0.6 \pm 0.2$ | $-0.8 \pm 0.3$ | $0.801 \pm 0.006$ | $0.97 \pm 0.01$ |
| **ECOSYSTEM** | | | | | | | |
| $\mathcal{S}_{\text{SIMP}}$ | 0.527 | 0.532 | 0.525 | $0.89 \pm 0.02$ | $0.92 \pm 0.03$ | $0.95 \pm 0.02$ | $0.93 \pm 0.03$ |
| $\mathcal{S}_{\text{CORR}}$ | 0.856 | 0.890 | 0.820 | $0.876 \pm 0.004$ | $0.877 \pm 0.004$ | $0.904 \pm 0.009$ | $0.888 \pm 0.003$ |
| $\mathcal{S}_{\text{HOMOL}}$ | 0.185 | 0.066 | 0.256 | $0.17 \pm 0.03$ | $0.09 \pm 0.04$ | $0.36 \pm 0.03$ | $0.33 \pm 0.03$ |
| $\mathcal{S}_{\text{PROC}}$ | 0.055 | 0.024 | 0.025 | $-0.01 \pm 0.03$ | $-0.1 \pm 0.05$ | $0.04 \pm 0.03$ | $0.08 \pm 0.05$ |
| $\mathcal{S}_{\text{DTW}}$ | 0.111 | 0.115 | 0.051 | $0.11 \pm 0.02$ | $0.05 \pm 0.03$ | $0.12 \pm 0.02$ | $0.15 \pm 0.03$ |
| $\mathcal{S}_{\text{NN}}$ | 0.133 | 0.133 | 0.146 | $0.304 \pm 0.005$ | $0.304 \pm 0.004$ | $0.30 \pm 0.03$ | $0.313 \pm 0.005$ |
| $\mathcal{S}_{\text{DIM}}$ | -0.882 | 0.60 | 0.664 | $0.38 \pm 0.08$ | $0.51 \pm 0.05$ | $0.90 \pm 0.02$ | $0.92 \pm 0.02$ |
| **TORUS** | | | | | | | |
| $\mathcal{S}_{\text{SIMP}}$ | 0.984 | 0.996 | 0.994 | $0.998 \pm 0.001$ | $0.999 \pm 0.001$ | $0.999 \pm 0.001$ | $0.998 \pm 0.002$ |
| $\mathcal{S}_{\text{CORR}}$ | 0.994 | 0.952 | 0.993 | $0.982 \pm 0.006$ | $0.87 \pm 0.03$ | $0.994 \pm 0.004$ | $0.99 \pm 0.01$ |
| $\mathcal{S}_{\text{HOMOL}}$ | 0.001 | -1.442 | -0.827 | $-0.6 \pm 0.06$ | $-0.4 \pm 0.2$ | $-0.3 \pm 0.2$ | $0.33 \pm 0.09$ |
| $\mathcal{S}_{\text{PROC}}$ | 0.157 | -0.102 | -0.008 | $0.1 \pm 0.1$ | $-0.07 \pm 0.08$ | $0.4 \pm 0.1$ | $0.4 \pm 0.1$ |
| $\mathcal{S}_{\text{DTW}}$ | 0.403 | 0.292 | 0.586 | $0.24 \pm 0.07$ | $0.19 \pm 0.07$ | $0.60 \pm 0.08$ | $0.50 \pm 0.09$ |
| $\mathcal{S}_{\text{NN}}$ | 0.269 | 0.194 | 0.444 | $0.28 \pm 0.03$ | $0.28 \pm 0.01$ | $0.42 \pm 0.01$ | $0.45 \pm 0.02$ |
| $\mathcal{S}_{\text{DIM}}$ | -0.619 | -0.652 | 0.722 | $0.1 \pm 0.1$ | $-0.3 \pm 0.3$ | $0.96 \pm 0.04$ | $0.71 \pm 0.01$ |
| **RÖSSLER** | | | | | | | |
| $\mathcal{S}_{\text{SIMP}}$ | 0.988 | 0.997 | 0.997 | $0.999 \pm 0.001$ | $0.997 \pm 0.001$ | $0.999 \pm 0.001$ | $0.999 \pm 0.001$ |
| $\mathcal{S}_{\text{CORR}}$ | 0.771 | 0.994 | 0.999 | $0.94 \pm 0.02$ | $0.87 \pm 0.03$ | $0.985 \pm 0.003$ | $0.997 \pm 0.003$ |
| $\mathcal{S}_{\text{HOMOL}}$ | 0.001 | 0.06 | 0.501 | $0.08 \pm 0.04$ | $0.08 \pm 0.07$ | $0.27 \pm 0.04$ | $0.55 \pm 0.07$ |
| $\mathcal{S}_{\text{PROC}}$ | 0.123 | -0.002 | 0.027 | $0.01 \pm 0.09$ | $0.33 \pm 0.04$ | $0.3 \pm 0.1$ | $0.25 \pm 0.06$ |
| $\mathcal{S}_{\text{DTW}}$ | 0.351 | 0.547 | 0.527 | $0.23 \pm 0.07$ | $0.43 \pm 0.05$ | $0.52 \pm 0.09$ | $0.62 \pm 0.05$ |
| $\mathcal{S}_{\text{NN}}$ | 0.332 | 0.742 | 0.762 | $0.43 \pm 0.03$ | $0.42 \pm 0.03$ | $0.64 \pm 0.01$ | $0.75 \pm 0.06$ |
| $\mathcal{S}_{\text{DIM}}$ | -0.48 | 0.423 | 0.727 | $0.64 \pm 0.04$ | $0.5 \pm 0.1$ | $0.694 \pm 0.05$ | $0.753 \pm 0.08$ |

[1] M. B. Kennel, R. Brown, and H. D. Abarbanel, Physical review A **45**, 3403 (1992).

[2] J. Huisman and F. J. Weissing, Nature **402**, 407 (1999).

[3] A. Asseman, T. Kornuta, and A. Ozcan, in *NIPS Modeling and Decision-making in the Spatiotemporal Domain Workshop* (2018).

[4] P. Laguna, R. G. Mark, A. Goldberg, and G. B. Moody, in *Computers in cardiology 1997* (IEEE, 1997) pp. 673–676.

[5] A. L. Goldberger, L. A. Amaral, L. Glass, J. M. Hausdorff, P. C. Ivanov, R. G. Mark, J. E. Mietus, G. B. Moody, C.-K. Peng, and H. E. Stanley, Circulation **101**, e215 (2000).

[6] S. S. Rangapuram, M. W. Seeger, J. Gasthaus, L. Stella, Y. Wang, and T. Januschowski, in *Advances in neural information processing systems* (2018) pp. 7785–7794.

[7] D. Dua and C. Graff, "UCI machine learning repository," (2017).

[8] R. Chaudhuri, B. Gercek, B. Pandey, A. Peyrache, and I. Fiete, Nature neuroscience **22**, 1512 (2019).

[9] S. L. Brunton, B. W. Brunton, J. L. Proctor, E. Kaiser, and J. N. Kutz, Nature communications **8**, 19 (2017).

[10] F. Pedregosa, G. Varoquaux, A. Gramfort, V. Michel, B. Thirion, O. Grisel, M. Blondel, P. Prettenhofer, R. Weiss, V. Dubourg, *et al.*, Journal of machine learning research **12**, 2825 (2011).

[11] M. P. Harrigan, M. M. Sultan, C. X. Hernández, B. E. Husic, P. Eastman, C. R. Schwantes, K. A. Beauchamp, R. T. McGibbon, and V. S. Pande, Biophysical journal **112**, 10 (2017).

[12] P. Virtanen, R. Gommers, T. E. Oliphant, M. Haberland, T. Reddy, D. Cournapeau, E. Burovski, P. Peterson, W. Weckesser, J. Bright, *et al.*, Nature Methods (2020).

[13] M. Abadi, P. Barham, J. Chen, Z. Chen, A. Davis, J. Dean, M. Devin, S. Ghemawat, G. Irving, M. Isard, *et al.*, in *12th {USENIX} Symposium on Operating Systems Design and Implementation ({OSDI} 16)* (2016) pp. 265–283.

[14] S. E. Otto and C. W. Rowley, SIAM Journal on Applied Dynamical Systems **18**, 558 (2019).

[15] C. Diks, W. Van Zwet, F. Takens, and J. DeGoede, Physical Review E **53**, 2169 (1996).

[16] H. Edelsbrunner and J. Harer, Contemporary mathematics **453**, 257 (2008).

[17] V. Venkataraman, K. N. Ramamurthy, and P. Turaga, in *2016 IEEE international conference on image processing (ICIP)* (IEEE, 2016) pp. 4150–4154.

[18] Q. H. Tran and Y. Hasegawa, Physical Review E **99**, 032209 (2019).

[19] C. Tralie, N. Saul, and R. Bar-On, The Journal of Open Source Software **3**, 925 (2018).

[20] N. Saul and C. Tralie, "Scikit-TDA: Topological Data Analysis for Python," (2019).

[21] G. Sugihara and R. M. May, Nature **344**, 734 (1990).

[22] G. Sugihara, R. May, H. Ye, C.-h. Hsieh, E. Deyle, M. Fogarty, and S. Munch, Science **338**, 496 (2012).

[23] S. Linderman, M. Johnson, A. Miller, R. Adams, D. Blei, and L. Paninski, in *Artificial Intelligence and Statistics* (2017) pp. 914–922.

[24] P. Grassberger and I. Procaccia, Physica D: Nonlinear Phenomena **9**, 189 (1983).

[25] M. Karl, M. Soelch, J. Bayer, and P. Van der Smagt, in *International Conference on Learning Representations* (2017) pp. 1–13.

[26] J. Durbin and S. J. Koopman, *Time series analysis by state space methods* (Oxford university press, 2012).

[27] E. Yemini, T. Jucikas, L. J. Grundy, A. E. Brown, and W. R. Schafer, Nature methods **10**, 877 (2013).

[28] T. N. Lal, T. Hinterberger, G. Widman, M. Schröder, N. J. Hill, W. Rosenstiel, C. E. Elger, N. Birbaumer, and B. Schölkopf, in *Advances in neural information processing systems* (2005) pp. 737–744.

[29] A. Bagnall, J. Lines, A. Bostrom, J. Large, and E. Keogh, Data Mining and Knowledge Discovery **31**, 606 (2017).

[30] H. Jiang and H. He, in *2017 International Joint Conference on Neural Networks (IJCNN)* (IEEE, 2017) pp. 3191–3198.