[Reviews · NeurIPS 2020]

Review 1

Summary and Contributions: This work considers the problem of reconstructing or analyzing a multidimensional time series governed by a deterministic dynamical system when one has access only to measurements from a single dimension of the time series. A broad literature in the physics community has explored this problem in recent decades, with much work focusing on methods that reconstruct the complete time series from a so-called delay embedding. The contribution of this paper is a method to perform this reconstruction using an autoencoder framework with a regularization technique inspired by a classical method in the physics literature, false nearest neighbors. The paper shows that the regularized autoencoder architecture performs better than a plain autoencoder and three baseline methods from the physics literature.

Strengths: The paper shows that bringing machine learning tools to a widely-used technique in time series analysis can bring performance benefits even in the presence of noise, giving this work the potential to impact the wide array of applications that use these methods. The method is appealingly simple, and the authors argue that it requires less parameter tuning than existing techniques. The method is shown to improve performance over comparison techniques on a broad range of datasets and metrics.

Weaknesses: Despite being an interesting observation and novel idea, the paper has a few key weaknesses both in the writing and in the work reported (detailed in the additional feedback below). In the writing, the technique is motivated as a general method for time series analysis, but is limited to systems governed by an underlying dynamical systems. This should be discussed in some detail, as this connection gets lost in the writing. Second, there are a number of relevant pieces of literature that the authors refer to indirectly but don’t cite explicitly (details below). Most importantly, the most related reference (ref. 35, which also uses an autoencoder to learn the relevant map) is only described with half a sentence. In the work itself, the simulation examples give a nice starting point, but are not comprehensive enough to be a convincing demonstration of the full claims of the paper. It’s not clear why the baselines used for comparisons were selected or if they are state of the art (see point 3 in additional feedback below). For example, as far as I can tell, the closely related ref. 35 is not used as a comparison method at all and it’s not clear why. Second, the authors sell their proposed method as requiring only “essentially one” parameter to be tuned, but there are many other parameters that should be considered tunable (see point 2 in additional feedback below). The authors must make a convincing case that the comparison methods are state of the art and that the method isn’t highly sensitive to the additional parameters discussed in point 2 in additional feedback.

Correctness: The paper is primarily an algorithmic paper. The development seems correct.

Clarity: Overall, the paper is very well written and organized. I believe the background and problem setup would benefit from a more detailed exposition on state space methods and delay embeddings.

Relation to Prior Work: The paper presents a thorough background on delay embedding methods (with the exception of Ref. 35, which seems to be the most closely-related method). The author mentions isometric versions of Whitney’s embedding theorem, but doesn’t cite some of the recent work in this area. I would include: R. G. Baraniuk and M. B. Wakin, Random Projections of Smooth Manifolds, Foundations of Computational Mathematics, vol. 9, no. 1, pp. 51-77, February 2009. A. Eftekhari and M. B. Wakin, New Analysis of Manifold Embeddings and Signal Recovery from Compressive Measurements, Applied and Computational Harmonic Analysis, vol. 39, no. 1, pp. 67-109, July 2015. K. L. Clarkson. Tighter bounds for random projections of manifolds. In Proc. Symp. Comput. Geom., pages 39–48. ACM, 2008. Similarly, the authors mention isometric (stable) versions of Takens’ theorem and should discuss the relationship to: A. Eftekhari, H. L. Yap, M. B. Wakin, and C. J. Rozell, Stabilizing Embedology: Geometry-Preserving Delay-Coordinate Maps, Physical Review E, vol. 97, no. 2, pp. 022222, February 2018.

Reproducibility: Yes

Additional Feedback: 1. The paper considers the setting in which the observed time series is governed by a dynamical system. However, when the problem is cast into a machine learning setup for general time series analysis, this distinction is sometimes lost. This may be a point to mention in the broader impacts section: in many applications it is not known if the time series data of interest is governed by a dynamical system. 2. The authors claim that a major advantage of the proposed method is that it requires less hyperparameter tuning than existing methods, only requiring the regularization parameter lambda and learning rate gamma to be set by the user. I have some concerns about this claim: a. Could the authors provide more evidence that the learning rate should not be considered a hyperparameter (“essentially one governing hyperparameter” in line 316)? After all, in lines 189-190 the learning rate is listed as a parameter that is tuned. b. In traditional state space methods, the number of lags between time samples in the delay map (tau in this paper) is considered a parameter to be tuned. It seems like this parameter could be considered "tuneable" in the proposed method, too, but the authors simply fix it to 1. c. In traditional methods the embedding dimension (d_E in this paper) is considered a parameter to be tuned. The contribution of this paper is an adaptation of the false nearest neighbors heuristic that is often used as a heuristic to select d_E. Since prior literature uses the same heuristic to select d_E, I do not understand why the embedding dimension should be considered a tunable parameter in other work but not here. d. On a related note, why should L not be considered a tunable parameter in the proposed method? The authors provide evidence that the proposed regularizer performs a type of variable selection that may reduce the importance of a perfect selection of L, but nonetheless it seems like the proper selection of this parameter may still influence the result. e. Since \bar{F}_m is recomputed for each batch, the technique could be sensitive to batch size. Which batch size is used and how was this selected? Does this significantly slow down processing? When computing the dimension in the results section how is \bar{F}_m computed? f. To support the claim that "[existing] embedding techniques are strongly sensitive to hyperparameter choice" (line 44), the authors cite only a paper that specifically discusses the convergent cross-mapping method, a specific application of delay embedding. 3. It’s not clear why the comparison techniques used were selected (and in fact, the abbreviations are not well-explained to even know clearly what they are). Can the authors provide some justification for their selection and whether they represent state of the art performance on the datasets/metrics considered? Also, the paper does not provide information on the computational cost of the proposed method compared to the comparison methods. Since the proposed method involves training an autoencoder this computational cost may be a significant disadvantage. 4. Since the main attractor reconstruction results in Figure 3 are on different scales and do not have colorbars, it is difficult to interpret the strength of the results without looking at the table in the appendix. For example, the results for the forecast metric look similar for the Lorenz and Torus systems, but Table S3 (in the appendix) shows that the proposed method is much better than the competitor method for the Lorenz system, while for the Torus system all methods have forecast accuracy > 0.98. The data should be presented with clear colorbars and consistent scales to make this apparent. 5. The authors note that “timescales and sampling rates are chosen so that the dominant Fourier peaks align” for the datasets used. This statement is not clear to me and should be explained more. The concern here is that data preprocessing may be critically important to the method (which appears to be true for some of the related methods we have worked with in this problem space). 6. The authors do not provide evidence for whether the heuristic used to select the parameter lambda (based on the results in Figure 5(B)) generalizes to other settings. 7. Minor points: a. The dimensionalities of the vectors and matrices discussed in Section 2 are a bit confusing and potentially inconsistent. In line 66 it seems like \bm{x}_t is a row vector, making X in line 73 Nxd_m. But \bm{x}_i in line 76 is a column vector, which would make the corresponding X a vector. And, if X is TxN (line 79) shouldn’t Y be LxN (line 81)? Also see Y and Yhat. It would also be helpful to define L before it is used on line 81. b. Section 2 could make the distinction between d, d_E, and L more clear. c. I’m unfamiliar with defining the dimension of a function g as in line 90 rather than defining the dimension of the output of g (i.e.\hat{Y}) d. S_dim is referred to on line 244 but it is never defined in the main text, only the appendix. e. References to Figure 3 are inconsistent in using A/B and top/bottom. f. In the Figure 4 caption, the last sentence mentions tau = 20 and xi = 0.5. Do these refer to the fixed values used for subfigures B and C? If so, the letters should be fixed in the last sentence. If not, further clarification of what that sentence means would be helpful. g. Figure 4B-C is incorrectly referred to as Figure 4A-B in several places. h. How is the true solution defined in Figure 5A? i. The symbols and labels given to the metrics used in Section 4.2 are confusing and inconsistent with the symbols and labels used in Figure 3 and in Section V of the appendix. j. In several places, "cross-map forecast accuracy" could be described more precisely. Is this measured by correlation coefficient? -------- Update after rebuttal: I am glad to see the addition of the autoencoder dimension, but it doesn't significantly change my opinion of the paper. Most importantly, there is significant confusion around the different dimensions being used in the manuscript. While I still recommend acceptance because I think this section could be removed entirely without harming the main results of the paper, the rebuttal missed an opportunity to clarify this point of confusion that was common to multiple reviewers. Given this, my score remains the same.


Review 2

Summary and Contributions: The paper presents a novel approach for reconstructing the underlying attractors of dynamical systems from partial (incomplete) observations of the system. The paper suggests to first augment the dimensionality using the method of lags/delays, and then reconstruct the data using a regularized autoencoder. Results are presented on multiple dynamical systems, both with known and unknown dynamics.

Strengths: The paper is well written and most parts are clearly described. The machine learning methodology is well presented, and the problem is relevant to the ML community. The experimental results are very extensive and promising. The discussion in the broader impact on privacy is very relevant, and raises an important concern if the underlying attractor can be inferred from partial observations.

Weaknesses: Overall, I like the methodology a lot, but as detailed in my comments, I am a bit concerned about the framing of the problem which might create some confusion, especially from the point of view of dynamical systems. The other aspect that needs improvement is the discussion on the different types of dimensionality introduced, as detailed below. 1) Many recent methods that try to recover an underlying attractor from partial observations (observation of only some part of the system) use the method of delays (aka, Takens embedding, time-lagged embedding, delay-coordinate maps). The current paper also uses this delay embedding (L76) but without making this connection clear. I find the two paragraphs L81-L110 fairly confusing (see also the comment below on dimensionality). a. L3-8: The current approach is presented in opposition to existing techniques that do dimension reduction. Many methods (Singular Spectrum Analysis, Nonlinear Laplacian Spectral Analysis, etc) that rely on a partial observation of the system, first embed these observations into a higher-dimensional space (via the method of delays) and then do some dimension reduction (via eigendecomposition of the covariance matrix, Laplacian method, Koopman operator, etc). The current paper does the same thing, first embeds the univariate observations via the method of delays (L76), and then uses an autoencoder to find a representation in a latent space (which can also be seen as a dimension reduction). b. L100: In practice, g is constructed from a composition between the time delays and often some eigendecomposition, not just the time delays. 2) There are many definitions of dimensionality (d, d_E, d_F, L, d_m, T) which creates some confusion: a. What is the connection between T and d_E? Are they both referring to the embedding dimension in the delay-coordinate embedding space, i.e., the number of delays/lags used to build the ``embedded’’ attractor? Here, I think there might also be a confusion around the notion of ``embedding space’’ as there are two spaces used: 1) the time-lagged embedding space, and 2) embedding space (latent space) of the autoencoder. b. In the end, I suppose an important goal is to find the true dimension of the dynamical system (d) (also Sect. 5.3). What is the connection between d and L ? L is chosen as the dimension of the inner layer of the autoencoder, and in experiments, L=10. But can we just choose L as large as possible, regardless of the intrinsic dimensionality d? For unknown attractors, is there a guarantee that L > d? Is there a connection between d and d_E? L163-164: ``true’’ embedding dimension d_E: this is confusing because I am not sure what d_E refers to exactly. If d_E is used as in L101 (equivalent to T in L76), is there a true (delay coordinate) embedding dimension? If d_E is similar to d as in L143, what is the difference between them?

Correctness: The method is correct, but some claims are not well explained. The empirical methodology is mostly well detailed, however there are too many evaluation metrics in my opinion without sufficient details to fully understand how well they work.

Clarity: The paper is well written and most parts are easy to follow, but there are some inconsistencies, some detailed below.

Relation to Prior Work: I believe this is the main shortcoming of the paper. There are many methods that have similar approaches, however these connections are not made clear in the paper (especially Sect. 2). Is there a difference between the method of lags that the authors use and the method of delays/time-lagged embedding from the state-of-the art?

Reproducibility: Yes

Additional Feedback: Other comments: 1. Why strange in the title? Does this approach only apply to strange attractors (i.e., they have a fractal structure)? Are all the real world datasets from the experiments generated by strange attractors, eg, neural spiking, old faithful, etc? 2. The methodology (L72-80) is presented for univariate time series (multidimensional time series are proposed as future work – L319). However the authors use “low-dimensional” often throughout the paper, but this can be higher than one-dimensional. The suggestion would be to either: 1) extend the methodology to multivariate data, or 2) state in the abstract (and elsewhere) that this is for univariate time series, not “low-dimensional”. 3. L21: incomplete observations is often referred in dynamical systems as “partial observations”. All that is available is some observation of the dynamical system, which need not be a direct observation of one of the states (as in the Lorenz example from the paper). 4. L27-28: recent – recently 5. L81: \theta subscript not used again 6. L84-86: The discussion about which type of similarity between Y and \hat{Y} is sought for is too brief and not clear, as this is a very important aspect. 7. L87 and L105: \mathcal{M} is the measurement operator, not the measurement. The measurement is x 8. L101: \tau is the timestep, similar to x_i in L76 (which is the discrete analog and has it’s own timestep). If I understand correctly the authors’ goal, d_E=T? 9. L121: [30] does not decompose the Hankel matrix, but a Laplacian matrix built from the Hankel matrix, and on top of that they do an SVD (which is not always needed). Singular Spectrum Analysis (SSA) [Ghil et al., 2002] decomposes directly the Hankel matrix. 10. Sect. 4.1: Fig. 2 is not mentioned in the text. It might be worthwhile mentioning that g is the encoder and g’ the decoder. 11. Sect. 4.2: How is the split between train-validation-test done? If this is done randomly, due to the Hankel matrix (time lags/delays) then one could have an important overlap of information between the three datasets. 12. L214: If L would be chosen much larger (especially for unknown attractors) what is the impact of augmenting the dimensionality be adding many zeros? 13. Fig. 3: there is no A and B and the similarity measures are slightly different than described in the text. What are the values in the similarity plots? The colorbar would be very useful. 14. Fig. 4: What does the negative forecast accuracy mean? How is the forecast accuracy defined exactly? Would be interesting for Fig. 4A to understand if using different observables, y1 vs y2 vs y3, leads to the same attractor. Caption: \tau and \eps_0 are for B and C, instead of A and B. 15. S_dim is not defined, but I assume it refers to the dimension similarity. 16. Fig. 5: What is the exact definition of dimensionality error? What does a dimensionality error of 2.5 mean? I would believe it should be integers, but I might be misunderstanding something. ---------------------------------------------------- Rebuttal: I would like to thank the authors for their responses. I have read the responses carefully, however unfortunately the distinction between the different dimensionalities used is still not clear to me. For example, it is not clear if the purpose is to learn d (the true number of ODEs, the intrinsic dimensionality) or d_E (the number of lags in line 101). Unfortunately the authors did not respond to some of my questions that are still open: • What is the connection between T and d_E? If \tau = 1, I suppose they are the same. In the response, line 54, the authors say d_E \approx d (not T). • Why strange in the title? Does this approach only apply to strange attractors (i.e., they have a fractal structure)? Are all the real world datasets from the experiments generated by strange attractors, eg, neural spiking, old faithful? (previous comment) • Is d_F needed? Which of the examples has a fractal structure? However, overall my biggest concern is the ambiguity between d and d_E. What is the main goal: to learn d or d_E? In line 9 of the response, d_E is reffered to as the “embedding dimension”, which I assume refers to the embedding dimension of the latent space of the autoencoder. But line 101 from the paper shows clearly that d_E is the number of lags, so the dimension of the delay-coordinate embedding, not of the latent space of the autoencoder. In the response, the author mention that the method and code have been extended to multivariate data. Are there any results added to the paper to show how the method is applied to multivariate data? I am not so much concerned by the quality of the results, as by the consistency between the method formulation and the results. If the method was modified to also apply to multivariate data, the results should contain at least one example on multivariate data. Small comments: • Response (b): “For unknown systems any large L can be used” – unfortunately I don’t fully agree with this statement. As the authors correctly state in their responses, the correct attractor is reconstructed “as long as L is larger than d”. But for unknown systems, and for very complex systems, there is no guarantee that the chosen L is larger than d. This remains an assumption (that the chosen L is larger than d), and it would be helpful to make this clear. • response (a): “L sets the maximum d_E expressible by the autoencoder” (line 9) or the maximum d (true number of ODEs)? What is the connection between d and d_E for unknown systems? • The authors find that d \approx d_E. Is this because all previous results were on univariate data? Is the same observed for multivariate data? For example, for a 100D system, if we observe data from 99 of the ODEs, do we still need d_E=100? Maybe it’s also from here the confusion between d and d_E. I do believe the paper offers a novel view of using autoencoders with delay embeddings and has a strong potential, however due to the inconsistencies - mainly related to the goal of learning d or d_E, and the exact definition of d_E (dimension of the latent space of the autoencoder or the number of lags), I will keep my previous score. I believe these things are central to the paper, and should be stated clearly to avoid any confusion.


Review 3

Summary and Contributions: This paper presents a method for reconstructing the attractor of a dynamical system when only low-dimensional time-series measurements of the system are available. The authors created attractors from time series by training an autoencoder on the Hankel measurement matrix for the system.

Strengths: Experiments show that the technique reconstructs the strange attractors of synthetic and real-world systems better than existing techniques, and that it creates consistent, predictive representations of different systems.

Weaknesses: There have been a lot work related to the state space reconstruction for time series. Several studies leverages singular-value decomposition of the Hankel matrix to reconstruct the state space. Thera have also been some works that used nonlinear kernel function to map time series into a high dimensional state space. This paper is a natural extension of previous work, which utilized autoencoder on the Hankel matrix to achieve state space reconstruction. The innovation of the paper is not significant.

Correctness: The claim and proposed method is correct in this paper.

Clarity: It is well written.

Relation to Prior Work: The work is different from previous contributions. But the innovation is not significant.

Reproducibility: Yes

Additional Feedback:


Review 4

Summary and Contributions: The paper shows how to reconstruct a dynamical system (attractor) from a low-dimensional measurements.

Strengths: The paper presents a method how to actually do this. Takens and Whitney describe and prove that this is possible, but do not provide a method how to actually achieve it. This paper does.

Weaknesses: The method is only demonstrated on three-dimensional systems. It is not clear how well the method scales.

Correctness: The methods are correct and the comparisons are valid.

Clarity: The paper is written very well.

Relation to Prior Work: Yes.

Reproducibility: Yes

Additional Feedback:

[Author Response · NeurIPS 2020]

We thank the four reviewers for their excitement about our work and detailed feedback! Overall, reviewers enjoyed
the problem and approach: **"potential to impact the wide array of applications" "appealingly simple," "novel ap-**
**proach", "very extensive and promising" "better than existing techniques"**; criticism centered around description
of prior work, and clarity of problem formulation and details. We address these concerns here
**(a) Regularizer design and $d_E$ (R2, R1).** We thank R2, and we hope that they will reconsider. We've revised the text
in order to clarify the problem framing and fix notational ambiguities and some typos, which we believe clarify R2's
major concerns (and a related point about $d_E$ by R1). $d$ is the number of independent dynamical variables (number of
ODEs); $d_F \leq d$ is attractor manifold dimension (non-integer for chaotic systems with fractal attractors; $d_F = 1$ for
limit cycles); $d_E$ is embedding dimension. The hyperparameter $L$ sets the maximum $d_E$ expressible by the autoencoder
(AE). However, the AE does not pick an integer $d_E$; rather, $d_E$ is estimated continuously post-training via the relative
variance of each latent variable averaged across train data (similar to PCA weights). Dimensionality score thus compares
explained variance (a function of latent index) between reconstruction and full dimensional system. Our experiments on
systems with known $d$ seek and find that $d_E \approx d$. Our regularizer takes a batch of latent activations, and estimates $\bar{F}_m$,
the proportion of new false neighbors indexed by number of latent dimensions. Since $\bar{F}_m$ is intensive, it only weakly
depends on batch size (we have added new physics references related to this observation).
**(b) Dependence on $L$ (R2, R1).** We have added new experiments showing that $L$ does not affect learned $d_E$ of latent
embedding as long as $L$ is larger than $d$ (see Fig H1); thus for unknown systems any large $L$ can be used.
**(c) Prior Work (R1, R3).** We performed new experiments and extended the main text discussion of Ref. 35, which
(to our knowledge) is the primary prior application of AE to attractor reconstruction. Ref. 35 embeds datasets via a
one-layer AE with $\tanh$ activation and MSE loss, similar to our existing baseline unregularized MLP (see appendix).
We performed a new set of experiments with a model exactly matching Ref. 35 (Fig H1), and find it is comparable to
our other baselines for the noisy prediction task (same task as ref. 35). We have added these results to the paper.
**(d) Higher dimensions (R4)** Thank you! We have clarified that the ecosystem results are 10D; pendulum is 4D. We've
added discussion and references to physics papers about mathematical limitations of embedding in the high-$d$ limit.
**(e) Existing work on state space modelling (R3):** Thank you, and we hope that you will reconsider. Our paper
doesn't claim to be the first AE applied to embedding (see (c) above). Indeed, $30\%$ of our submission is a literature
review of state space modelling (SSM); there we demonstrate several clear areas of novelty: (1) Our paper focuses on a
**fully novel loss function and regularizer** rooted in the classical theory of dynamical systems, and we shows that this
regularizer strongly constrains and improves AE representations, in contrast to prior AEs used for SSM. (2) Our paper
uses a variety of **novel measures of attractor fidelity**—e.g. topology, neighbor coverage, fractal dimension—which go
beyond few-timestep RMSE forecasting errors (the primary metric in previous works). (3) We show strong results for
embedding **consistency across replicates, and robustness to Brownian stochasticity** (a more complex noise source
than the measurement errors studied in prior works), two desirable embedding properties not explored previously.
**R1 & R2 additional comments:** Thank you so much for
detailed feedback; we've fixed all wording, framing, and
added suggested references; we regret that space limits us
to major concerns not covered above: **R1 Misc: 8.1.2a,b**
We revised hyperparameter discussion to add nuance: we
mean that our experiments achieve strong results only by
varying learning rate and regularizer strength, and the for-
mer is only tuned to ensure that train loss plateaus. Rather
than pre-select embedding timescale, we favor fixing large
$T$ and batch size, and letting the AE learn how much to
weigh different timepoints. **8.1.2c,d,e** See (a) above. **8.3**
We've moved details from appendix 5 into main text. There

Figure H1: (A, B) Updated Fig 4 with ref. 35 baseline (yellow). (C) Similar activity patterns in first 10 latent units for fnn-AEs trained with $L = 10, 20, 30, 40, 50$ (blue to red). (Inset) dimension accuracy vs $L$ also shows no dependence.

are few widely-used definitions of attractor similarity (many SSM papers from ML authors focus on prediction, not
verisimilitude, and many physics papers are qualitative), and so we report multiple established and novel metrics in
order to give a holistic view. We've added the caveat that FNN-AE is more expensive than ETD/tICA, especially on
small datasets, but only marginally more expensive than unregularized AE. **8.5** We revised to clarify that time series are
Fourier-resampled only to ensure consistency across datasets; preprocessing/filtering otherwise has little effect (hence
noise results) **8.6** We re-ran experiments and confirmed with pendulum data; we will add this to the appendix.
**R2 Misc: 3.2, 8.8, 8.12, 8.14-16** See (a, b); we've also moved Appendix 5 details to main text to clarify scoring
metrics. We always refer to size $L$ latent space as embedding space: $T$ time delays only serve as a featurization of input
to the AE, which seeks (and we find achieves) $d_E \approx d$ (not $T$). $d_E \leq L$ is computed *post-hoc* from relative latent
activations by finding variance of the $L$ latent variables across train set, giving continuous measure of dimensionality
(thus unaffected by zero-padding). Thus $d_E$ is neither a hyperparameter nor a direct AE output. **R2.8.2** The method
and the code we're releasing now works for multivariate time series; we will highlight this. **R2.8.11** No leakage; when
available, we use 2 different datasets or initial conditions; otherwise we use first $N$ and last $N$ points of a time series
with length $\gg 2N$. **R2.8.16** We scale lower bound to mean, not theoretical min (see Nassar et al ICLR 2019, Eq. 25).

[Meta-Review · NeurIPS 2020]

This paper uses a novel regularizer based on false neighbors to find latent delay embedding. There were intense discussions among the reviewers on sec 2. I tend to agree with R1 who puts more emphasis on the innovation of the approach. However, I found the important details of the method is hidden in the supplement. I suggest switching parts of sec 2 with the details about the method from the supplementary text.